# Supervised Pretraining Can Learn In-Context Reinforcement Learning

**Jonathan N. Lee**[*1]     **Annie Xie**[*1]     **Aldo Pacchiano**[2 3]     **Yash Chandak**[1]

**Chelsea Finn**[1]     **Ofir Nachum**     **Emma Brunskill**[1]

[1]Stanford University [2]Broad Institute of MIT and Harvard [3]Boston University

## Abstract

Large transformer models trained on diverse datasets have shown a remarkable ability to *learn in-context*, achieving high few-shot performance on tasks they were not explicitly trained to solve. In this paper, we study the in-context learning capabilities of transformers in decision-making problems, i.e., reinforcement learning (RL) for bandits and Markov decision processes. To do so, we introduce and study the *Decision-Pretrained Transformer* (*DPT*), a supervised pretraining method where a transformer predicts an optimal action given a query state and an in-context dataset of interactions from a diverse set of tasks. While simple, this procedure produces a model with several surprising capabilities. We find that the trained transformer can solve a range of RL problems in-context, exhibiting both exploration online and conservatism offline, despite not being explicitly trained to do so. The model also generalizes beyond the pretraining distribution to new tasks and automatically adapts its decision-making strategies to unknown structure. Theoretically, we show DPT can be viewed as an efficient implementation of Bayesian posterior sampling, a provably sample-efficient RL algorithm. We further leverage this connection to provide guarantees on the regret of the in-context algorithm yielded by DPT, and prove that it can learn faster than algorithms used to generate the pretraining data. These results suggest a promising yet simple path towards instilling strong in-context decision-making abilities in transformers.

## 1   Introduction

For supervised learning, transformer-based models trained at scale have shown impressive abilities to perform tasks given an input context, often referred to as few-shot prompting or in-context learning [1]. In this setting, a pretrained model is presented with a small number of supervised input-output examples in its context, and is then asked to predict the most likely completion (i.e. output) of an unpaired input, without parameter updates. Over the last few years, in-context learning has been applied to solve a range of tasks [2] and a growing number works are beginning to understand and analyze in-context learning for supervised learning [3, 4, 5, 6]. In this work, our focus is to study and understand in-context learning applied to sequential decision-making, specifically in the context of reinforcement learning (RL) settings. Decision-making (e.g. RL) is considerably more dynamic and complex than supervised learning. Understanding and leveraging in-context learning here could potentially unlock significant improvements in an agent's ability to adapt and make few-shot decisions in response to observations from the world. Such capabilities are instrumental for practical applications ranging from robotics to recommendation systems.

---

*Equal contribution. Correspondence to `jnl@stanford.edu` and `anniexie@stanford.edu`. Code is available at `https://github.com/jon-lee/decision-pretrained-transformer`

For in-context decision-making [7, 8, 9], rather than input-output tuples, the context takes the form of state-action-reward tuples representing a dataset of interactions with an unknown environment. The agent must leverage these interactions to understand the dynamics of the world and what actions lead to good outcomes. A hallmark of good decision-making in online RL algorithms is a judicious balance of selecting exploratory actions to gather information and selecting increasingly optimal actions by exploiting that information [10]. In contrast, an RL agent with access to only a suboptimal offline dataset should produce a policy that conservatively selects actions [11]. An ideal in-context decision-maker should exhibit similar behaviors.

To study in-context decision-making formally, we propose a new simple supervised pretraining objective, namely, to train (via supervised learning) a transformer to predict an optimal action label[2] given a query state and an in-context dataset of interactions, across a diverse set of tasks. We refer to the pretrained model as a Decision-Pretrained Transformer (DPT). Once trained, DPT can be deployed as either an online or offline RL algorithm in a new task by passing it an in-context dataset of interactions and querying it for predictions of the optimal action in different states. For example, online, the in-context dataset is initially empty and DPT's predictions are uncertain because the new task is unknown, but it fills the dataset with its interactions as it learns and becomes more confident about the optimal action. We show empirically and theoretically that DPT yields a surprisingly effective in-context decision-maker with regret guarantees. As it turns out, DPT effectively performs posterior sampling — a provably sample-efficient Bayesian RL algorithm that has historically been limited by its computational burden [12, 13]. We summarize our main findings below.

- **Predicting optimal actions alone gives rise to near-optimal decision-making algorithms.** The DPT objective is solely based on predicting optimal actions from in-context interactions. At the outset, it is not immediately apparent that these predictions at test-time would yield good decision-making behavior when the task is unknown and behaviors such as online exploration are necessary to solve it. Intriguingly, DPT as an algorithm is capable of dealing with this uncertainty in-context. For example, despite not being explicitly trained to explore, DPT exhibits an exploration strategy on par with hand-designed algorithms, as a means to discover the optimal actions.

- **DPT generalizes to new decision-making problems, offline and online.** We show DPT can handle reward distributions unseen in its pretraining data on bandit problems as well as unseen goals, dynamics, and datasets in simple MDPs. This suggests that the in-context strategies learned during pretraining are robust and generalizable without any parameter updates at test time.

- **DPT improves over the data used to pretrain it by exploiting latent structure.** As an example, in parametric bandit problems, specialized algorithms can leverage structure (such as linear rewards) and offer provably better regret. However, a representation must be known in advance. Perhaps surprisingly, we find that pretraining on linear bandit problems, even with unknown representations, leads DPT to select actions and explore in a way that matches an efficient linear bandit algorithm. This holds even when the source pretraining data comes from a suboptimal algorithm (i.e., one that does not take advantage of any latent structure), demonstrating the ability to learn improved in-context strategies beyond what it was trained on.

- **Posterior sampling can be implemented via in-context learning.** Posterior sampling (PS), a generalization of Thompson Sampling, can provably sample-efficiently solve online RL problems [12, 13], but a common criticism is the lack of computationally efficient ways to update and sample from a posterior distribution. DPT can be viewed as learning a posterior distribution over optimal actions, shortcutting the PS procedure. Under some conditions, we show theoretically that DPT in-context is equivalent to PS. Furthermore, DPT's prior and posterior updates are grounded in data rather than needing to be specified *a priori*. This suggests that in-context learning could be a possible path towards scalable and practical RL via posterior sampling.

## 2   Related Work

**Meta-learning.**   Algorithmically, in-context learning falls under the meta-learning framework [14, 15]. At a high-level, these methods attempt to learn some underlying shared structure of the training distribution of tasks to accelerate learning of new tasks. For decision-making and RL, there is often

---

[2]If not explicitly known, the optimal action can be determined by running any (potentially inefficient) minimax-optimal regret algorithm for each pretraining task.

a choice in what shared 'structure' is specifically learned such as the dynamics of the task [16, 17, 18, 19], a task context identifier [20, 21, 22, 23], temporally extended skills and options [24, 25, 26], or initialization of a neural network policy [27, 28]). In-context learning can be viewed as taking a more agnostic approach by learning the learning algorithm itself, more similar to [29, 30, 31, 32, 8]. Algorithm Distillation (AD) [7, 33] falls under this category, using autoregressive supervised learning to distill traces of a single-task RL algorithm into a task-agnostic model. While DPT also leverages autoregressive SL, it does not distill an existing RL algorithm in order to imitate how to learn. Instead, we train DPT to predict task-specific optimal actions, yielding potentially emergent strategies at test time that automatically leverage the task structure to behave similarly to posterior sampling.

**Autoregressive transformers for decision-making.** In decision-making fields such as RL and imitation learning, transformer models trained using autoregressive supervised action prediction have proliferated [34], inspired by the successes of these techniques for large language models [35, 36, 1]. For example, Decision Transformer (DT) [37, 38, 8] uses a transformer to autoregressively model sequences of actions from offline experience data, conditioned on the achieved return. During inference, one can then query the model conditioned on a desired return value. This approach has been shown to scale favorably to large models and multi-task settings [39], at times exceeding the performance of large-scale multi-task imitation learning with transformers [40, 41, 42]. However, DT is known to be provably (and unboundedly) sub-optimal in common scenarios [43, 44]. A common criticism of DT, and supervised learned transformers in general, is their inability to improve upon the dataset. For example, there is little reason for DT to output meaningful behavior if conditioned on return higher than any observed in training, without strong extrapolation assumptions [43]. In contrast, a major contribution of our work is theoretical and empirical evidence for the ability of DPT to improve over behaviors seen in the dataset in terms of regret.

**Value and policy-based offline RL.** Offline RL algorithms offer the opportunity to learn from existing datasets. To address distributional shift, many prior algorithms incorporate the principle of value pessimism [45, 46, 47, 48], or policy regularization [49, 50, 51, 52, 53]. To reduce the amount of offline data required in a new task, methods for offline meta-RL can reuse interactions collected in a set of related tasks [54, 55, 56]. However, they still must address distribution shift, requiring solutions such as policy regularization [54] or additional online interactions [57]. DPT follows the success of autoregressive models like DT and AD, avoiding these issues. With our pretraining objective, DPT also leverages offline datasets for new tasks more effectively than AD.

## 3 In-Context Learning Model

**Basic decision models.** The basic decision model of our study is the finite-horizon Markov decision process (MDP). An MDP is specified by the tuple $\tau = \langle \mathcal{S}, \mathcal{A}, T, R, H, \rho \rangle$ to be solved, where $\mathcal{S}$ is the state space, $\mathcal{A}$ is the action space, $T : \mathcal{S} \times \mathcal{A} \to \Delta(\mathcal{S})$ is the transition function, $R : \mathcal{S} \times \mathcal{A} \to \Delta(\mathbb{R})$ is the reward function, $H \in \mathbb{N}$ is the horizon, and $\rho \in \Delta(\mathcal{S})$ is the initial state distribution. A learner interacts with the environment through the following protocol: (1) an initial state $s_1$ is sampled from $\rho$; (2) at time step $h$, the learner chooses an action $a_h$ and transitions to state $s_{h+1} \sim T(\cdot|s_h, a_h)$, and receives a reward $r_h \sim R(\cdot|s_h, a_h)$. The episode ends after $H$ steps. A policy $\pi$ maps states to distributions over actions and can be used to interact with the MDP. We denote the optimal policy as $\pi^\star$, which maximizes the value function $V(\pi^\star) = \max_\pi V(\pi) := \max_\pi \mathbb{E}_\pi \sum_h r_h$. When necessary, we use the subscript $\tau$ to distinguish $V_\tau$ and $\pi^\star_\tau$ for the specific MDP $\tau$. We assume the state space is partitioned by $h \in [H]$ so that $\pi^\star$ is notationally independent of $h$. Note this framework encompasses multi-armed bandit settings where the state space is a single point, e.g. $\mathcal{S} = \{1\}$, $H = 1$, and the optimal policy is $a^\star \in \operatorname{argmax}_{a \in \mathcal{A}} \mathbb{E}[r_1|a_1 = a]$.

**Pretraining.** We give pseudocode in Algorithm 1 and a visualization of the transformer in Figure 1. Details on the precise structure of the transformer can be found in Section A.1. Let $\mathcal{T}_{\text{pre}}$ be a distribution over tasks at the time of pretraining. A task $\tau \sim \mathcal{T}_{\text{pre}}$ can be viewed as a specification of an MDP, $\tau = \langle \mathcal{S}, \mathcal{A}, T, R, H, \rho \rangle$. The distribution $\mathcal{T}_{\text{pre}}$ can span different reward and transition functions and even different state and action spaces. We then sample a context (or a prompt) which consists of a dataset $D \sim \mathcal{D}_{\text{pre}}(\cdot; \tau)$ of interactions between the learner and the MDP specified by $\tau$. $D = \{s_j, a_j, s'_j, r_j\}_{j \in [n]}$ is a collection of transition tuples taken in $\tau$. We refer to $D$ as the *in-context dataset* because it provides the contextual information about $\tau$. $D$ could be generated through variety of means, such as: (1) random interactions within $\tau$, (2) demonstrations from an expert, and (3) rollouts of an algorithm. Additionally, we independently sample a query state $s_{\text{query}}$ from the distribution $\mathcal{D}_{\text{query}}$ over states $\mathcal{S}$ and a label $a^\star$ is sampled from the optimal policy $\pi^\star_\tau(\cdot|s_{\text{query}})$

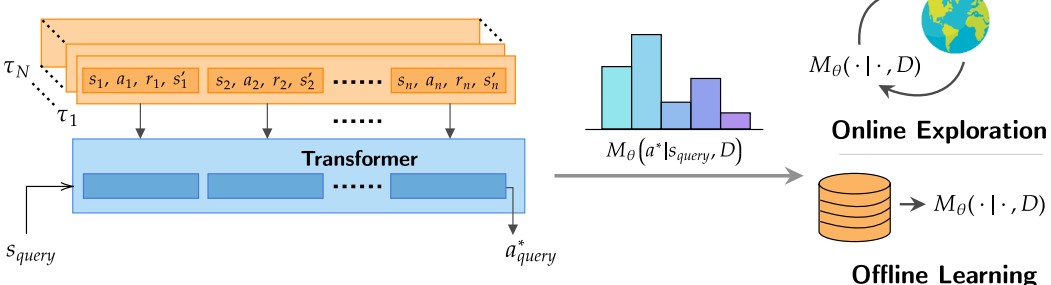

Figure 1: A transformer model $M_\theta$ is pretrained to predict an optimal action $a^\star_{\text{query}}$ from a state $s_{\text{query}}$ in a task, given a dataset of interactions from that task. The resulting Decision-Pretrained Transformer (DPT) learns a distribution over the optimal action conditioned on an in-context dataset. $M_\theta$ can be deployed in *new* tasks online by collecting data on the fly or offline by immediately conditioning on a static dataset.

---

**Algorithm 1** Decision-Pretrained Transformer (DPT): Training and Deployment

---

1: // Collecting pretraining dataset
2: Initialize empty pretraining dataset $\mathcal{B}$
3: **for** $i$ in $[N]$ **do**
4:     Sample task $\tau \sim \mathcal{T}_{\text{pre}}$, in-context dataset $D \sim \mathcal{D}_{\text{pre}}(\cdot; \tau)$, query state $s_{\text{query}} \sim \mathcal{D}_{\text{query}}$
5:     Sample label $a^\star \sim \pi^\star_\tau(\cdot|s_{\text{query}})$ and add $(s_{\text{query}}, D, a^\star)$ to $\mathcal{B}$
6: **end for**
7: // Pretraining model on dataset
8: Initialize model $M_\theta$ with parameters $\theta$
9: **while** not converged **do**
10:     Sample $(s_{\text{query}}, D, a^\star)$ from $\mathcal{B}$ and predict $\hat{p}_j(\cdot) = M_\theta(\cdot|s_{\text{query}}, D_j)$ for all $j \in [n]$
11:     Compute loss in (2) with respect to $a^\star$ and backpropagate to update $\theta$.
12: **end while**
13: // Offline test-time deployment
14: Sample unknown task $\tau \sim \mathcal{T}_{\text{test}}$, sample dataset $D \sim \mathcal{D}_{\text{test}}(\cdot; \tau)$
15: Deploy $M_\theta$ in $\tau$ by choosing $a_h \in \text{argmax}_{a \in \mathcal{A}} M_\theta(a|s_h, D)$ at step $h$
16: // Online test-time deployment
17: Sample unknown task $\tau \sim \mathcal{T}_{\text{test}}$ and initialize empty $D = \{\}$
18: **for** ep in max_eps **do**
19:     Deploy $M_\theta$ by sampling $a_h \sim M_\theta(\cdot|s_h, D)$ at step $h$
20:     Add $(s_1, a_1, r_1, \ldots)$ to $D$
21: **end for**

---

for task $\tau$ (see Section 5.3 for how to implement this in common practical scenarios). We denote the joint pretraining distribution over tasks, in-context datasets, query states, and action labels as $P_{pre}$:

$$P_{pre}(\tau, D, s_{\text{query}}, a^\star) = \mathcal{T}_{\text{pre}}(\tau)\mathcal{D}_{\text{pre}}(D; \tau)\mathcal{D}_{\text{query}}(s_{\text{query}})\pi^\star_\tau(a^\star|s_{\text{query}}). \quad (1)$$

Given the in-context dataset $D$ and a query state $s_{\text{query}}$, we can train a model to predict the optimal action $a^\star$ in response simply via supervised learning. Let $D_j = \{(s_1, a_1, s'_1, r_1), \ldots, (s_j, a_j, s'_j, r_j)\}$ denote the partial dataset up to $j$ samples. Formally, we aim to train a causal GPT-2 transformer model $M$ parameterized by $\theta$, which outputs a distribution over actions $\mathcal{A}$, to minimize the expected loss over samples from the pretraining distribution:

$$\min_\theta \mathbb{E}_{P_{pre}} \sum_{j \in [n]} \ell\left(M_\theta(\cdot \mid s_{\text{query}}, D_j), a^\star\right). \quad (2)$$

Throughout, we set the loss to be the negative log-likelihood with $\ell(M_\theta(\cdot \mid s_{\text{query}}, D_j), a^\star) := -\log M_\theta(a^\star \mid s_{\text{query}}, D_j)$. This framework can work for both discrete and continuous $\mathcal{A}$. For our experiments with discrete $\mathcal{A}$, we use a softmax parameterization for the distribution of $M_\theta$, essentially treating this as a classification problem. The resulting output model $M_\theta$ can be viewed as an algorithm that takes in a dataset of interactions $D$ and can be queried with a forward pass for predictions of the optimal action via inputting a query state $s_{\text{query}}$. We refer to the trained model $M_\theta$ as a Decision-Pretrained Transformer (DPT).

**Testing.**    After pretraining, a new task (MDP) $\tau$ is sampled from a test-task distribution $\mathcal{T}_{\text{test}}$. If the DPT is to be tested *offline*, then a dataset (prompt) is a sampled $D \sim \mathcal{D}_{\text{test}}(\,\cdot\,;\tau)$ and the policy that the model in-context learns is given conditionally as $M_\theta(\cdot \mid \cdot, D)$. Namely, we evaluate the policy by selecting action $a_h \in \text{argmax}_a M_\theta(a|s_h, D)$ when the learner visits state $s_h$. If the model is to be tested *online* through multiple episodes of interaction, then the dataset is initialized as empty $D = \{\}$. At each episode, $M_\theta(\cdot \mid \cdot, D)$ is deployed where the model samples $a_h \sim M_\theta(\cdot|s_h, D)$ upon observing state $s_h$. Throughout a full episode, it collects interactions $\{s_1, a_1, r_1, \ldots, s_H, a_H, r_H\}$ which are subsequently appended to $D$. The model then repeats the process with another episode, and so on until a specified number of episodes has been reached. A key distinction of the testing phase is that there are no updates to the parameters of $M_\theta$. This is in contrast to hand-designed RL algorithms that would perform parameter updates or maintain statistics using $D$ to learn from scratch. Instead, the model $M_\theta$ performs a computation through its forward pass to generate a distribution over actions conditioned on the in-context $D$ and query state $s_h$.

## 4    Learning in Bandits

We begin with an empirical investigation of DPT in a multi-armed bandit, a well-studied special case of the MDP where the state space $\mathcal{S}$ is a singleton and the horizon $H = 1$ is a single step. We will examine the performance of DPT both when aiming to select a good action from offline historical data and for online learning where the goal is to maximize cumulative reward from scratch. Offline, it is critical to account for uncertainty due to noise as certain actions may not be sampled well enough. Online, it is critical to judiciously balance exploration and exploitation to minimize overall regret. For detailed descriptions of the experiment setups, see Appendix A.

**Pretraining distribution.**    For the pretraining task distribution $\mathcal{T}_{\text{pre}}$, we sample 5-armed bandits ($|\mathcal{A}| = 5$). The reward function for arm $a$ is a normal distribution $R(\cdot|s, a) = \mathcal{N}(\mu_a, \sigma^2)$ where $\mu_a \sim \text{Unif}[0, 1]$ independently and $\sigma = 0.3$. To generate in-context datasets $\mathcal{D}_{\text{pre}}$, we randomly generate action frequencies by sampling probabilities from a Dirichlet distribution and mixing them with a point-mass distribution on one random arm (see details in Appendix A.3). Then we sample the actions accordingly from this distribution. This encourages diversity of the in-context datasets. The optimal policy $\pi_\tau^\star$ for bandit $\tau$ is $\text{argmax}_a \mu_a$, which we can easily compute during pretraining. We pretrain the model $M_\theta$ to predict $a^\star$ from $D$ as described in Section 3 for datasets up to size $n = 500$.

**Comparisons.**    We compare to several well-known algorithms for bandits[3]. All of the algorithms are designed to reason in a particular way about uncertainty based on their observations.

- Empirical mean algorithm (Emp) selects the action with the highest empirical mean reward naively.
- Upper Confidence Bound (UCB) selects the action with the highest upper confidence bound.
- Lower Confidence Bound (LCB) selects the action with the highest lower confidence bound.
- Thompson Sampling (TS) selects the action with the highest sampled mean from a posterior distribution over reward models. The prior and likelihood functions are Gaussian.

Emp and TS [58, 59] can both be used for offline or online learning; UCB [60] is an online algorithm that achieves exploration via optimism under uncertainty; and LCB [61, 62] is used to minimize suboptimality given an offline dataset by selecting actions pessimistically. We evaluate algorithms with standard bandit metrics. Offline, we use the suboptimality $\mu_{a^\star} - \mu_{\hat{a}}$ where $\hat{a}$ is the chosen action. Online, we use cumulative regret: $\sum_k \mu_{a^\star} - \mu_{\hat{a}_k}$ where $\hat{a}_k$ is the $k$th action chosen.

**DPT learns to reason through uncertainty.**    As shown in Figure 2a, in the offline setting, DPT significantly exceeds the performance of Emp and LCB while matching the performance of TS, when the in-context datasets are sampled from the same distribution as during pretraining. The results suggest that the transformer is capable of reasoning through uncertainty caused by the noisy rewards in the dataset. Unlike Emp which can be fooled by noisy, undersampled actions, the transformer has learned to *hedge* to a degree. However, it also suggests that this hedging is fundamentally different from what LCB does, at least on this specific distribution[4].

Interestingly, the same transformer produces an extremely effective online bandit algorithm when sampling actions instead of taking an argmax. As shown in Figure 2b, DPT matches the performance

---

[3]See Appendix A.2 for additional details such as hyperparameters.

[4]Note our randomly generated environments are equally likely to have expert-biased datasets and adversarial datasets, so LCB is not expected to outperform here [61].

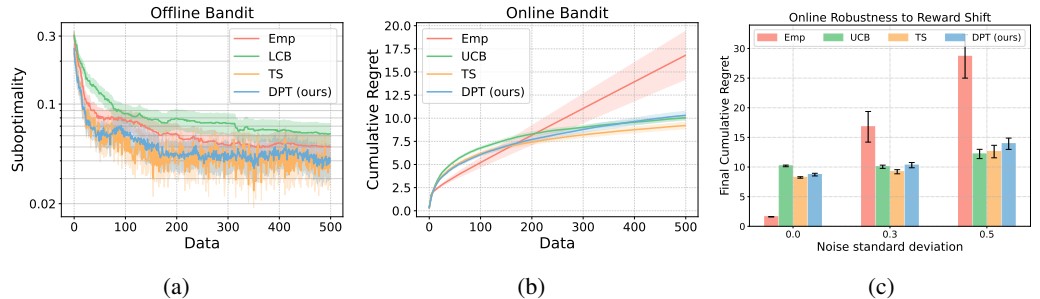

Figure 2: (a) Offline performance on in-distribution bandits, given random in-context datasets. (b) Online cumulative regret on bandits. (c) Final (after 500 steps) cumulative regret on out-of-distribution bandits with different Gaussian noise standard deviations. The mean and standard error are computed over 200 test tasks.

of classical optimal algorithms, UCB and TS, which are specifically designed for exploration. This is notable because DPT was not explicitly trained to explore, but its emergent strategy is on par with some of the best. In Figure 2c, we show this property is robust to noise in the rewards not seen during pretraining by varying the standard deviation. In Appendix B, we show this generalization happens offline too and even with unseen Bernoulli rewards.

**Leveraging latent structure from suboptimal data.** We now investigate whether DPT can learn to leverage the inherent structure of a problem class, even without prior knowledge of this structure and even when learning from in-context datasets that do not explicitly utilize it. More precisely, we consider $\mathcal{T}_{\text{pre}}$ to be a distribution over *linear* bandits, where the reward function is given by $\mathbb{E}\left[r \mid a, \tau\right] = \langle \theta_\tau, \phi(a) \rangle$ and $\theta_\tau \in \mathbb{R}^d$ is a task-specific parameter vector and $\phi : \mathcal{A} \to \mathbb{R}^d$ is fixed feature vector that is the same for all tasks. Given the feature representation $\phi$, LinUCB [63], a UCB-style algorithm that leverages $\phi$, should achieve regret $\widetilde{\mathcal{O}}(d\sqrt{K})$ over $K$ steps, a substantial gain over UCB and TS when $d \ll |\mathcal{A}|$. We do not assume $\phi$ is known to DPT, besides whatever can be gleaned from the pretraining data. To evaluate the ability of DPT to uncover latent structure, we collect pretraining data with in-context datasets gathered by standard TS, which does not leverage the linear structure. Figures 3a and 3b show that DPT can exploit the unknown linear structure, implicitly learning a surrogate for $\phi$, allowing to do more informed exploration online and decision-making offline. It is nearly on par with LinUCB (which is given $\phi$). We also see that DPT significantly outperforms the source of its pretraining data, TS which did not know or use the latent structure to gather the in-context datasets. These results present evidence that (1) DPT can automatically leverage structure, and (2) supervised learning-based approaches to RL *can* potentially learn novel exploration strategies that transcend the quality of their pretraining data.

**Adapting to expert-biased datasets.** A common assumption in offline RL is that datasets tend to be a mixture between optimal data (e.g. expert demonstrations) and suboptimal data (e.g. random interactions) [64]. Hence, LCB is generally effective in practice and the pretraining and testing distributions should be biased towards this setting. Motivated by this, we pretrain another DPT model where $\mathcal{D}_{\text{pre}}$ is generated by mixing the in-context datasets with varying fractions of expert data, biasing $\mathcal{D}_{\text{pre}}$ towards datasets that contain more examples of the optimal action. We denote this model by DPT-Exp. In Figure 3c, we plot the test-time performance of both pretrained models when evaluated on new offline datasets with varying percentages of expert data[5]. Our results suggest that when the pretraining distribution is also biased towards expert-suboptimal data, DPT-Exp behaves similarly to LCB, while DPT continues to resemble TS. This is quite interesting as, for other methods such as TS, it is less clear how to automatically incorporate the right amount of expert bias to yield the same effect, but DPT can leverage this from pretraining.

## 5 Learning in Markov Decision Processes

We next study how DPT can tackle Markov decision processes by testing its ability to perform exploration and credit assignment. In the following experiments, the DPT demonstrates generalization to new tasks, scalability to image-based observations, and capability to stitch in-context behaviors (Section 5.2). This section also examines whether DPT can be pretrained with datasets and action labels generated by a different RL algorithm, rather than the exact optimal policy (Section 5.3).

---

[5]That is, $0\%$ is fully random while $100\%$ has only optimal actions in the in-context dataset.

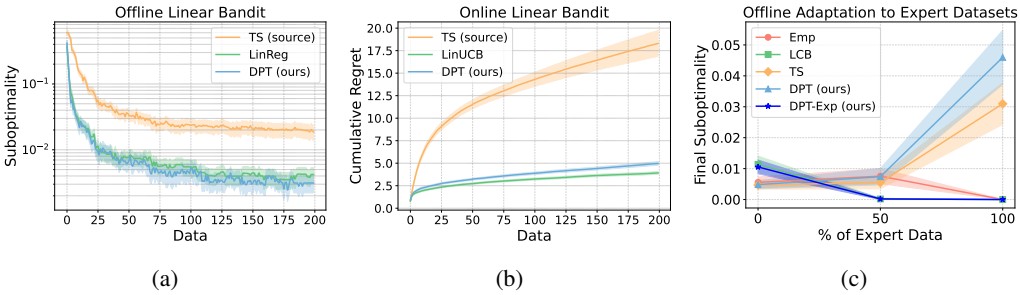

Figure 3: (a) Offline performance of DPT trained on linear bandits from TS source data. LinReg does linear regression and outputs the greedy action. (b) Online cumulative regret of the same model. The mean and standard error are computed over 200 test tasks. (c) Offline performance on expert-biased datasets. DPT pretrained on a different prior continues to match TS, but DPT-Exp trained from a more representative prior excels.

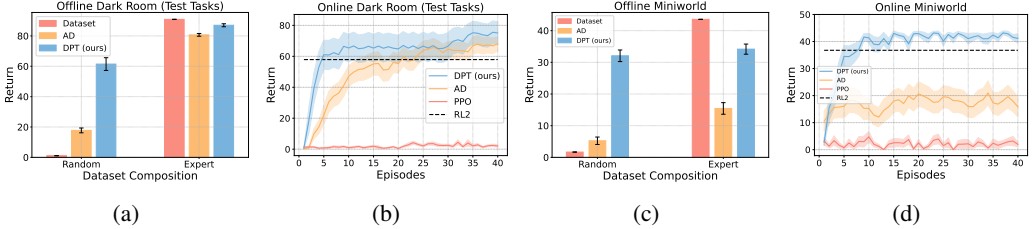

Figure 4: (a) Offline performance on held-out Dark Room goals, given random and expert datasets. (b) Online performance on held-out Dark Room goals. (c) Offline performance on Miniworld (images), given random and expert datasets. (d) Online performance on Miniworld (images) after 40 episodes. We report the average and standard error of the mean over 100 different offline datasets in (a) and (c) and 20 online trials in (b) and (d).

## 5.1 Experimental Setup

**Environments.** We consider environments that require targeted exploration to solve the task. The first is Dark Room [22, 7], a 2D discrete environment where the agent must locate the unknown goal location in a $10 \times 10$ room, and only receives a reward of $1$ when at the goal. We hold out a set of goals for generalization evaluation. The second is Miniworld [65], a 3D visual navigation problem to test the scalability of DPT to image observations. The agent is in a room with four boxes of different colors, and must find the target box. The target color is unknown initially. It receives a reward of $1$ only when near the correct box. Environments and pretraining dataset details are in App. A.4 and A.5.

**Comparisons.** Our experiments aim to understand the effectiveness of DPT by comparing it to other context-based meta-RL algorithms based on supervised and RL objectives.

- Proximal Policy Optimization (PPO) [66]: We compare to this single-task RL algorithm, which trains from scratch without any pretraining data, to contextualize the performance of DPT and other meta-RL algorithms.
- Algorithm Distillation (AD) [7]: AD first generates a dataset of learning histories by running an RL algorithm in each training task. Then, given a sampled subsequence $h_j = (s_j, a_j, r_j, \ldots, s_{j+c})$ from a learning history, a transformer is trained to predict the next action $a_{j+c}$ from the history.
- RL$^2$ [29]: This online meta-RL comparison uses a recurrent neural network to adapt the agent's policy from the given context. Unlike AD and DPT, which are trained with a supervised objective, the RL$^2$ agent is trained to maximize the expected return with PPO.

PPO and RL$^2$ are online algorithms, while AD is capable of learning both offline and online. Details on the implementation of these algorithms can be found in Appendix A.2.

## 5.2 Main Results

**Generalizing to new offline datasets and tasks.** To study the generalization capabilities of DPT, we evaluate the model in Dark Room on a set of $20$ held-out goals not in the pretraining dataset. When given an expert dataset, DPT achieves near-optimal performance. Even when given a random dataset, which has an average total reward of $1.1$, DPT obtains a much higher average return of

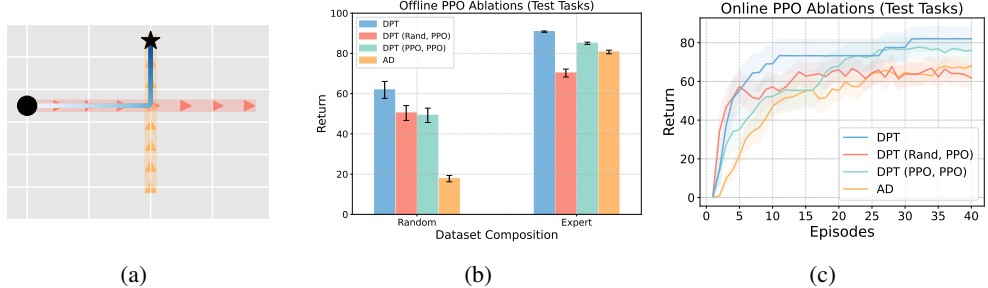

(a)                    (b)                    (c)

Figure 5: (a) In *Dark Room (Three Tasks)*, DPT stitches a new, optimal trajectory to the goal (blue) given two in-context demonstrations of other tasks (pink and orange). (b) Offline Dark Room performance of DPT trained on PPO data. (c) Online Dark Room performance of DPT trained on PPO data.

$61.5$ (see Fig. 4a). Qualitatively, we observe that when the in-context dataset contains a transition to the goal, DPT immediately exploits this and takes a direct path to the goal. In contrast, while AD demonstrates strong offline performance with expert data, it performs worse in-context learning with random data compared to DPT. The difference arises because AD is trained to infer a better policy than the in-context data, but not necessarily the optimal one.

We next evaluate DPT, AD, $RL^2$, and PPO online without any prior data from the $20$ test-time Dark Room tasks, shown in Fig. 4b. After $40$ episodes, PPO does not make significant progress towards the goal, highlighting the difficulty of learning from such few interactions alone. $RL^2$ is trained to perform adaptation within four episodes each of length $100$, and we report the performance after the four adaptation episodes. Notably, DPT on average solves each task faster than AD and reaches a higher final return than $RL^2$, demonstrating its capability to explore effectively online even in MDPs. In Appendix B, we also present results on generalization to new dynamics.

**Learning from image-based observations.** In Miniworld, the agent receives RGB image observations of $25 \times 25$ pixels. As shown in Fig. 4d, DPT can solve this high-dimensional task offline from both random and expert datasets. Compared to AD and $RL^2$, DPT also learns online more efficiently.

**Stitching novel trajectories from in-context subsequences.** A desirable property of some offline RL algorithms is the ability to stitch suboptimal subsequences from the offline dataset into new trajectories with higher return. To test whether DPT exhibits stitching, we design the *Dark Room (Three Tasks)* environment in which there are three possible tasks. The pretraining data consists only of expert demonstrations of two of them. At test-time DPT is evaluated on third unseen task, but its offline dataset is only expert demonstrations of the original two. Despite this, it leverages the data to infer a path solving the third task (see Fig. 5a).

### 5.3 Learning from Algorithm-Generated Policies and Rollouts

So far, we have only considered action labels provided by an optimal policy. However, in some tasks, an optimal policy is not readily available even in pretraining. In this experiment, we use actions labeled by a policy learned via PPO and in-context datasets sampled from PPO replay buffers. We train PPO agents in each of the $80$ train tasks for $1K$ episodes to generate $80K$ total rollouts, from which we sample the in-context datasets. This variant, DPT (PPO, PPO), performs on par with DPT and still better than AD, as shown in Figures 5b and 5c. DPT (PPO, PPO) can be viewed as a direct comparison between our pretraining objective and that of AD, given the same pretraining data but just used differently. We also evaluated a variant, DPT (Rand, PPO), which pretrains on random in-context datasets (like DPT), but still using PPO action labels. The performance is worse than other DPT variants in some settings, but only marginally so. In Appendix B, we analyze the sensitivity of DPT to other hyperparameters, such as context size and amount of pretraining data. These results give further evidence that DPT can learn an algorithm that transcends its pretraining data.

## 6 Theory

We now shed light on the observations of the previous empirical results through a theoretical analysis. Our main result shows that DPT (under a slight modification to pretraining) essentially is capable of executing posterior sampling (PS) reinforcement learning in-context. PS is a generalization of

Thompson Sampling for RL in MDPs [12]. It maintains and samples from a posterior over tasks $\tau$ given historical data $D$ and executes optimal policies $\pi_\tau^\star$ (see Appendix C for a formal outline). It is provably sample-efficient with online Bayesian regret guarantees [13], but maintaining posteriors is generally computationally intractable. The ability for DPT to perform PS in-context suggests a path towards computation- and provably sample-efficient RL with priors learned from the data.

## 6.1 History-Dependent Pretraining and Assumptions

We start with a modification to the pretraining of DPT. Rather than conditioning only on $s_\text{query}$ and $D$ to predict $a^\star \sim \pi_\tau^\star(\cdot|s_\text{query})$, we propose also conditioning on a sequence $\xi_h = (s_{1:h}, a_{1:h}^\star)$ where $s_{1:h} \sim \mathfrak{S}_h \in \Delta(\mathcal{S}^h)$ is a distribution over sets of states, independent of $\tau$, and $a_{h'}^\star \sim \pi_\tau^\star(\cdot|s_{h'})$ for $h' \in [h]$. Thus, we use $\pi_\tau^\star$ to label both the query state (which is the prediction label) and the sequence of states sampled from $\mathfrak{S}_h$. Note that this does not require any environment interactions and hence no sampling from either $T_\tau$ or $R_\tau$. At test-time at step $h$, this will allow us to condition on the history $\xi_{h-1}$ of states that $M_\theta$ visits and the actions that it takes in those states. Formally, the learned $M_\theta$ is deployed as follows, given $D$. (1) At $h = 0$, initialize $\xi_0 = ()$ to be empty. (2) At step $h$, visit $s_h$ and find $a_h$ by sampling from $M_\theta(\cdot|s_\text{query}, D, \xi_{h-1})$. (3) Append $(s_h, a_h)$ to $\xi_{h-1}$ to get $\xi_h$. Note for bandits and contextual bandits ($H = 1$), there is no difference between this and the original pretraining procedure of prior sections because $\xi_0$ is empty. For MDPs, the original DPT can be viewed as a practically convenient approximation.[6]

We now make several assumptions to simplify the analysis. First, assume $\mathcal{D}_\text{query}$, $\mathcal{D}_\text{pre}$, and $\mathfrak{S}$ have sufficient support such that all conditional probabilities of $P_{pre}$ are well defined. Similar to other studies of in-context learning [67], we assume $M_\theta$ fits the pretraining distribution exactly with enough coverage and data, so that the focus of the analysis is just the in-context learning abilities.

**Assumption 1.** *(Learned model is consistent). Let $M_\theta$ denote the pretrained model. For all $(s_{query}, D, \xi_h)$, we have $P_{pre}(a|s_{query}, D, \xi_h) = M_\theta(a|s_{query}, D, \xi_h)$ for all $a \in \mathcal{A}$.*

To provide some cursory justification, if $M_\theta$ is the global minimizer of (2), then we might expect that $\mathbb{E}_{P_{pre}}\|P_{pre}(\cdot|s_\text{query}, D, \xi_h) - M_\theta(\cdot|s_\text{query}, D, \xi_h)\|_1^2 \to 0$ as the number of pretraining samples $N \to \infty$ with high probability for transformer model classes of bounded complexity (see Proposition C.1). Approximate versions of the above assumptions are easily possible but obfuscate the key elements of the analysis. We also assume that the in-context dataset $D \sim \mathcal{D}_\text{pre}$ is compliant [62], meaning that the actions from $D$ can depend only on the observed history and not additional confounders. Note that this still allows $\mathcal{D}_\text{pre}$ to be very general — it could be generated randomly or from adaptive algorithms like PPO or TS. Without compliance $\mathcal{D}_\text{pre}$ can influence $M_\theta$. In Proposition 6.4, we show that all compliant $\mathcal{D}_\text{pre}$ form a sort of equivalence class that generate the same $M_\theta$. For the remainder, we assume all $\mathcal{D}_\text{pre}$ are compliant.

**Definition 6.1** (Compliance). *The in-context dataset distribution $\mathcal{D}_{pre}(\cdot; \tau)$ is compliant if, for all $i \in [n]$, the $i$th action of the dataset, $a_i$, is conditionally independent of $\tau$ given the $i$th state $s_i$ and partial dataset, $D_{i-1}$, so far. In other words, the distribution $\mathcal{D}_{pre}(a_i|s_i, D_{i-1}; \tau)$ is invariant to $\tau$.*

## 6.2 Main Results

**Equivalence of DPT and PS.** We now state our main result which shows that the trajectories generated by a pretrained $M_\theta$ will follow the same distribution as those from a well-specified PS algorithm. In particular, let PS use the well-specified prior $\mathcal{T}_\text{pre}$. Let $\tau_c$ be an arbitrary task. Let $P_\text{ps}(\cdot \mid D, \tau_c)$ and $P_{M_\theta}(\cdot \mid D, \tau_c)$ denote the distributions over trajectories $\xi_H \in (\mathcal{S} \times \mathcal{A})^H$ generated from running PS and $M_\theta(\cdot|\cdot, D, \cdot)$, respectively, in task $\tau_c$ given historical data $D$.

**Theorem 1** (DPT $\iff$ PS). *Let the above assumptions hold. Then, $P_\text{ps}(\xi_H \mid D, \tau_c) = P_{M_\theta}(\xi_H \mid D, \tau_c)$ for all trajectories $\xi_H$.*

**Regret implications.** With almost no extra work besides verifying assumptions, we can see this result in action. Let us specialize to the finite MDP setting [13]. Suppose we pretrain $M_\theta$ on a distribution $\mathcal{T}_\text{pre}$ over MDPs with $S := |\mathcal{S}|$ and $A := |\mathcal{A}|$. Let $\mathcal{D}_\text{pre}$ be constructed by uniform sampling $(s_i, a_i)$ and observing $(r_i, s_i')$ for $i \in [KH]$. Let $\mathbb{E}[r_h|s_h, a_h] \in [0, 1]$. And let $\mathcal{D}_\text{query}$ and $\mathfrak{S}_h$ be uniform over $\mathcal{S}$ and $\mathcal{S}^h$ (for all $h$) respectively. Finally, let $\mathcal{T}_\text{test}$ be the distribution over test

---

[6]It is still possible to implement this version, but it leaves some undesirable design choices to be made.

tasks with the same cardinalities. For a task $\tau$, define the online cumulative regret of DPT over $K$ episodes as $\text{Reg}_\tau(M_\theta) := \sum_{k \in [K]} V_\tau(\pi_\tau^\star) - V_\tau(\hat{\pi}_k)$ where $\hat{\pi}_k(\cdot|s_h) = M_\theta(\cdot|s_h, D_{(k-1)}, \xi_{h-1})$ and $D_{(k)}$ contains the first $k$ episodes collected from $\hat{\pi}_{1:k}$.

**Corollary 6.2** (Finite MDPs). *Suppose that $\sup_\tau \mathcal{T}_{test}(\tau)/\mathcal{T}_{pre}(\tau) \leq \mathcal{C}$ for some $\mathcal{C} > 0$. For the above MDP setting, the pretrained model $M_\theta$ satisfies $\mathbb{E}_{\mathcal{T}_{test}}[\text{Reg}_\tau(M_\theta)] \leq \widetilde{\mathcal{O}}(\mathcal{C}H^{3/2}S\sqrt{AK})$.*

A similar analysis due to [68] allows us to see theoretically why pretraining on (latently) linear bandits can lead to an in-context algorithm competitive with linear bandit algorithms, even when the in-context datasets are generated by algorithms unaware of this structure. We observed this empirically in Section 4. Consider a similar setup as there where $\mathcal{S}$ is a singleton, $\mathcal{A}$ is finite but large, $\theta_\tau \in \mathbb{R}^d$ is sampled as $\theta_\tau \sim \mathcal{N}(0, I/d)$, $\phi : \mathcal{A} \to \mathbb{R}^d$ is a fixed feature map with $\sup_{a \in \mathcal{A}} \|\phi(a)\|_2 \leq 1$, and the reward of $a \in \mathcal{A}$ in task $\tau$ is distributed as $\mathcal{N}(\langle \theta_\tau, \phi(a) \rangle, 1)$. This time, we let $\mathcal{D}_{\text{pre}}(\cdot; \tau)$ be given by running Thompson Sampling with Gaussian priors and likelihood functions on $\tau$.

**Corollary 6.3** (Latent representation learning in linear bandits). *For $\mathcal{T}_{test} = \mathcal{T}_{pre}$ in the above linear bandit setting, $M_\theta$ satisfies $\mathbb{E}_{\mathcal{T}_{test}}[\text{Reg}_\tau(M_\theta)] \leq \widetilde{\mathcal{O}}(d\sqrt{K})$.*

This significantly improves over the $\widetilde{\mathcal{O}}(\sqrt{|\mathcal{A}|K})$ upper regret bound for TS that does not leverage the linear structure. This highlights how DPT can have provably tighter upper bounds on future bandit problems than the algorithms used to generate its (pretraining) data. Note that if there is additional structure in the tasks which yields a tighter regret bound (for example if there are only a small finite number of MDPs in the possible distribution), that may further improve performance, such as by removing the dependence on the state, action, or dimension in these cases.

**Invariance of $M_\theta$ to compliant $\mathcal{D}_{\text{pre}}$.** Our final result sheds light on how $\mathcal{D}_{\text{pre}}$ impacts the final DPT behavior $M_\theta$. Combined with Assumption 1, $M_\theta$ is invariant to $\mathcal{D}_{\text{pre}}$ satisfying Definition 6.1.

**Proposition 6.4.** *Let $P_{pre}^1$ and $P_{pre}^2$ be pretraining distributions that differ only by their in-context dataset distributions, denoted by $\mathcal{D}_{pre}^1$ and $\mathcal{D}_{pre}^2$. If $\mathcal{D}_{pre}^1$ and $\mathcal{D}_{pre}^2$ are compliant with the same support, then $P_{pre}^1(a^\star|s_{query}, D, \xi_h) = P_{pre}^2(a^\star|s_{query}, D, \xi_h)$ for all $a^\star, s_{query}, D, \xi_h$.*

That is, if we generate in-context datasets $D$ by running various algorithms that depend only on the observed data in the current task, we will end up with the same $M_\theta$. For example, TS could be used for $\mathcal{D}_{\text{pre}}^1$ and PPO for $\mathcal{D}_{\text{pre}}^2$. Expert-biased datasets discussed in Section 4 violate Definition 6.1, since privileged knowledge of $\tau$ is being used. This helps explain our empirical results that pretraining on expert-biased datasets leads to a qualitatively different learned model at test-time (Fig. 3c).

## 7 Discussion

We studied the problem of in-context RL. We introduced a new pretraining method and transformer model, DPT, which is trained via supervised learning to predict optimal actions given an in-context dataset of interactions. Through in-depth evaluations in classic decision problems in bandits and MDPs, we showed that this simple objective naturally gives rise to an in-context RL algorithm that is capable of online exploration and offline decision-making in a way that leverages structure in the pretraining data. In some cases, the RL algorithm learned can transcend the quality of the pretraining data. Our empirical and theoretical results provide first steps towards understanding these capabilities and what factors are important for success. Via the simplicity of pretraining, we can sidestep the complexities of hand-designing exploration or conservatism in RL algorithms while simultaneously allowing the transformer to derive novel few-shot strategies. These findings underscore the potential of supervised pretraining to equip transformer models with in-context RL abilities.

**Limitations and future work.** One limitation of DPT is the requirement of optimal actions at pretraining. Empirically, we find that this requirement can be relaxed by using actions generated by another RL-trained agent during pretraining, which only leads to a slight loss in performance. However, fully understanding this problem and how best to leverage multi-task decision-making datasets remains a key open problem. We also discussed that the practical implementation for MDPs differs from true posterior sampling. It would be interesting to further understand and bridge this empirical-theoretical gap in the future. Finally, further investigation is required to understand the implications of these findings for existing transformer-based large language models that are increasingly being deployed in decision-making settings [69].

## Acknowledgments and Disclosure of Funding

We thank Yu Bai, Anirudh Goyal, Akshay Krishnamurthy, Evan Liu, Subhojyoti Mukherjee, Tung Nguyen, Lucy Shi, Ben Van Roy, and Sherry Yang for helpful discussions, feedback, and suggestions on related work. We also thank anonymous reviewers for their careful and constructive feedback. This work was supported in part by NSF grant 2112926 and ONR grant N00014-21-1-2685. JNL acknowledges support from the NSF GRFP. AP would like to thank the support of the Eric and Wendy Schmidt Center at the Broad Institute of MIT and Harvard. This work was supported in part by funding from the Eric and Wendy Schmidt Center at the Broad Institute of MIT and Harvard.

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

## Additional Related Work

**In-context learning.** Beyond decision-making and reinforcement learning, our approach takes inspiration from general in-context learning, a phenomenon observed most prominently in large language models in which large-scale autoregressive modelling can surprisingly lead to a model that exhibits meta-learning capabilities [1]. Recently, there has been great interest in understanding the capabilities and properties of in-context learning in various settings [4, 70, 5, 71, 72, 73, 74, 75, 6, 76, 77, 78]. While a common hypothesis suggests that this phenomenon is due to properties of the data used to train large language models [3], our work suggests that this phenomenon can also be encouraged in general settings via adjustments to the pre-training objective. In fact, DPT could be interpreted as explicitly encouraging the ability to perform Bayesian inference, which is a popular explanation for the mechanism behind in-context learning for large language models [67]. For meta-learning in-context, work by [79] proposed Transformer Neural Processes to address problems traditionally addressed with neural processes. Their work focuses on supervised learning-style input-output pairs for meta-learning, but they also applied this to contextual bandits to learn rewards.

For decision-making, in addition to the works discussed in the main paper, several works have applied transformers for in-context RL. AdA [32] meta-learns a transformer online (like RL$^2$). A base online RL algorithm is used to optimize the transformer, which uses online access to the simulator. The authors also use a curriculum learning method to improve the learning process. DPT is learned from offline supervised pretraining and it is agnostic to how the pretraining data is acquired. We believe many of the ideas in AdA are complementary and can be ported to DPT. Prompt-DT [8] is designed for offline RL and conditions on expert demos and uses return-to-go in the context much like the Decision Transformer [37]. This necessitates expert demos as input at test-time. DPT is meant for both online and offline in-context RL and does not require test-time demos (but it can benefit from them e.g. Fig. 4a). DPT does not use return-to-go, and instead predicts optimal actions from arbitrary in-context interaction datasets.

**Posterior Sampling.** Posterior sampling originates from the seminal work of [59], and has been popularized and thoroughly investigated in recent years by a number of authors [58, 80, 12, 13, 81, 68]. For bandits, it is often referred to as Thompson Sampling, but the framework is easily generalizable to RL, as shown in [12, 13]. The principle is as follows: begin with a prior over possible models (i.e. reward and transition functions), and maintain a posterior distribution over models by updating as new interactions are made. At decision-time, sample a model from the posterior and execute its optimal policy. The aforementioned prior works have developed strong theoretical guarantees on Bayesian and frequentist regret for posterior sampling. Despite its desirable theoretical characteristics, a major limitation is that computing the posterior is often computationally intractable. Recently, the community has made efforts towards scalable posterior sampling often by leveraging approximations [82, 83, 84, 85]. In Section 6, we show that a version of the DPT model learned from pretraining can be viewed as implementing posterior sampling without specifying complicated priors or posterior updates. Instead, the posterior update is implicitly learned through pretraining to predict the optimal action. This suggests that in-context learning via supervised pretraining (or meta-learning more generally) could be a promising alternative approach towards scalable posterior sampling.

On the theoretical side, [19] show it is possible to meta-learn prior distribution parameters for Thompson Sampling (TS) using a meta-exploration policy. The resulting TS algorithm can then be deployed on new problems. This is potentially consistent with the observation that DPT can approximate the right prior from offline data.

## A   Implementation and Experiment Details

### A.1   DPT Architecture: Formal Description

In this section, we provide a detailed description of the architecture alluded to in Section 3 and Figure 1. See hyperparameter details for models in their respective sections. The model is implemented in Python with PyTorch [86]. The backbone of the transformer architecture we use is an autoregressive GPT-2 model from the HuggingFace `transformers` library.

For the sake of exposition, we suppose that $\mathcal{S}$ and $\mathcal{A}$ are subsets of $\mathbb{R}^{d_S}$ and $\mathbb{R}^{d_A}$ respectively. We handle discrete state and action spaces with one-hot encoding. Consider a single training datapoint

---
**Algorithm 2** Decision-Pretrained Transformer (detailed)
---
1: `// Collecting pretraining dataset`
2: Initialize empty dataset $\mathcal{B}$
3: **for** $i$ in $[N]$ **do**
4:     Sample task $\tau \sim \mathcal{T}_{\text{pre}}$
5:     Sample interaction dataset $D \sim \mathcal{D}_{\text{pre}}(\cdot; \tau)$ of length $n$
6:     Sample $s_{\text{query}} \sim \mathcal{D}_{\text{query}}$ and $a^{\star} \sim \pi^{\star}_{\tau}(\cdot|s_{\text{query}})$
7:     Add $(s_{\text{query}}, D, a^{\star})$ to $\mathcal{B}$
8: **end for**
9: `// Training model on dataset`
10: Initialize model $M_{\theta}$ with parameters $\theta$
11: **while** not converged **do**
12:     Sample $(s_{\text{query}}, D, a^{\star})$ from $\mathcal{B}$
13:     Predict $\hat{p}_j(\cdot) = M_{\theta}(\cdot|s_{\text{query}}, D_j)$ for all $j \in [n]$.
14:     Compute loss in (5) with respect to $a^{\star}$ and backpropagate to update $\theta$.
15: **end while**
---

---
**Algorithm 3** Offline test-time deployment (detailed)
---
1: `// Task and offline dataset are generated without learner's control`
2: Sample unknown task $\tau \sim \mathcal{T}_{\text{test}}$
3: Sample dataset $D \sim \mathcal{D}_{\text{test}}(\cdot; \tau)$
4: `// Deploying offline policy` $M_{\theta}(\cdot|\cdot, D)$
5: $s_1 = \texttt{reset}(\tau)$
6: **for** $h$ in $[H]$ **do**
7:     $a_h = \text{argmax}_{a \in \mathcal{A}} M_{\theta}(\cdot|s_h, D)$ `// Most likely action`
8:     $s_{h+1}, r_h = \texttt{step}(\tau, a_h)$
9: **end for**
---

---
**Algorithm 4** Online test-time deployment (detailed)
---
1: `// Online, dataset is empty as learning is from scratch`
2: Initialize $D = \{\}$
3: Sample unknown task $\tau \sim \mathcal{T}_{\text{test}}$
4: **for** `ep` in `max_eps` **do**
5:     $s_1 = \texttt{reset}(\tau)$
6:     **for** $h$ in $[H]$ **do**
7:         $a_h \sim M_{\theta}(\cdot|s_h, D)$ `// Sample action from predicted distribution`
8:         $s_{h+1}, r_h = \texttt{step}(\tau, a_h)$
9:     **end for**
10:     `// Experience from previous episode added to dataset`
11:     Add $(s_1, a_1, r_1, \ldots)$ to $D$
12: **end for**
---

derived from an (potentially unknown) task $\tau$: we have a dataset $D$ of interactions within $\tau$, a query state $s_{\text{query}}$, and its corresponding optimal action $a^\star = \pi_\tau^\star(s_{\text{query}})$. We construct the embeddings to be passed to the GPT-2 backbone in the following way. From the dataset $D = \{(s_j, a_j, s'_j, r_j)\}_{j \in [n]}$, we construct vectors $\xi_j = (s_j, a_j, s'_j, r_j)$ by stacking the elements of the transition tuple into dimension $d_\xi := 2d_S + d_A + 1$ for each $j$ in the sequence. This sequence of $n$ elements is concatenated with another vector $v := (s_{\text{query}}, \mathbf{0})$ where the $\mathbf{0}$ vector is a vector of zeros of sufficient length to make the entire element dimension $d_\xi$. The $(n+1)$-length sequence is given by $X = (v, \xi_1, \ldots, \xi_n)$. As order does not often matter for the dataset $D^7$, we do not use positional encoding in order to take advantage of this invariance. We first apply a linear layer $\texttt{Linear}(X)$ and pass the result to the transformer, which outputs the sequence $Y = (\hat{y}_0, \hat{y}_1, \ldots, \hat{y}_n)$. In the continuous action case, these can be used as is for predictions of $a^\star$. For the discrete action case, we use them as logits to be converted to either a distribution over actions in $\mathcal{A}$ or one-hot vector predictions of $a^\star$. Here, we compute action probabilities

$$\hat{p}_j = \texttt{softmax}(\hat{y}_j) \in \Delta(\mathcal{A}) \tag{3}$$

Because of the GPT-2 causal architecture (we defer details to the original papers [87, 1]), we note that $\hat{p}_j$ depends only on $s_{\text{query}}$ and the partial dataset $D_j = \{(s_k, a_k, s'_k, r_k)\}_{k \in [j]}$, which is why we write the model notation,

$$M_\theta(\cdot | s_{\text{query}}, D_j) = \hat{p}_j(\cdot), \tag{4}$$

to denote that the predicted probabilities of the $j$th element only depend on $D_j$ and not the entire $D$ for the model $M$ with parameters $\theta \in \Theta$. For example, with $j = 0$, the prediction of $a^\star$ is made without any contextual information about the task $\tau$ except for $s_{\text{query}}$, which can be interpreted as the prior over $a^\star$. We measure loss of this training example via the cross entropy for each $j \in [n]$:

$$-\sum_{j \in [n]} \log \hat{p}_j(a^\star) \tag{5}$$

**Intuition.** Elements of the inputs sequence $X$ represent transitions in the environment. When passed through the GPT-2 transformer, the model learns to associate elements of the sequence via the standard query-key-value mechanism of the attention model. The query state $s_{\text{query}}$ is demarcated by its zeros vector (which also acts as padding). Unlike other examples of transformers used for decision-making such as the Decision Transformer [37] and Algorithm Distillation [7], DPT does not separate the individual $(s, a, s', r)$ into their own embeddings to be made into one long sequence. This is because we view the transition tuples in the dataset as their own singletons, to be related with other singletons in the dataset through the attention mechanism. We note that there are various other implementation variations one could take, but we found success and robustness with this one.

### A.2 Implementation Details

#### A.2.1 Bandit algorithms

First, we describe the comparisons from the bandit experiments with hyperparameters.

**Empirical Mean (Emp).** Emp has no hyperparameters, but we give it some mechanism to avoid degenerate scenarios. In the offline setting, Emp will only choose from actions that have at least one example in the dataset. This gives Emp and LCB-style effect when actions are missing. Similarly, online, Emp will sample each action at least once before defaulting to its real strategy. These changes only improve Emp.

**Upper Confidence Bound (UCB).** According to the Hoeffding bound, we choose actions as $\hat{a} \in \text{argmax}_{a \in \mathcal{A}} \left\{ \hat{\mu}_a + \sqrt{1/n_a} \right\}$ where $\hat{\mu}_a$ is the empirical mean so far for action $a$ and $n_a$ is the number of times $a$ has been chosen so far. To arrive at this constant for the bonus, we coarsely tried a set of plausible values given the noise and found this to perform the best.

**Lower Confidence Bound (LCB).** We choose actions as $\hat{a} \in \text{argmax}_{a \in \mathcal{A}} \left\{ \hat{\mu}_a - \sqrt{1/n_a} \right\}$ where $\hat{\mu}_a$ is the empirical mean so far for action $a$ and $n_a$ is the number of times $a$ has been chosen so far.

---

[7]This is not always true such as when data comes from an algorithm such as PPO or Thompson Sampling.

**Thompson Sampling (TS).** Since the means are sampled uniformly from $[0, 1]$, Gaussian TS is partially misspecified; however, we set prior mean and variance to $\frac{1}{2}$ and $\frac{1}{12}$ to match the true ones. The noise model was well-specified with the correct variance. In the linear experiments of Figure 3a and Figure 3b, we set the prior mean and variance to 0 and 1 to fit the true ones better.

**LinUCB.** We choose $\hat{a}_t \in \text{argmax}_{a \in \mathcal{A}} \langle \hat{\theta}_t, \phi(a) \rangle + \beta \|\phi(a)\|_{\hat{\Sigma}_t^{-1}}$ where $\beta = 1$ and $\hat{\Sigma}_t = I + \sum_{s \in [t-1]} \phi(a_s)\phi(a_s)^\top$ and $\hat{\theta}_t = \hat{\Sigma}_t^{-1} \sum_{s \in [t-1]} r_s \phi(a_s)$. Here, $r_s$ and $a_s$ are the reward and action observed at time $s$.

**LinReg.** LinReg (offline) is the same as LinUCB except we set $\beta = 0$ to greedily choose actions.

**DPT.** The transformer for DPT has an embedding size of 32, context length of 500 for basic bandits and 200 for linear bandits, 4 hidden layers, and 4 attention heads per attention layer for all bandits. We use the AdamW optimizer with weight decay 1e-4, learning rate 1e-4, and batch-size 64. For all experiments, we shuffle the in-context dataset $D$ since order does not matter except in the linear bandit.

### A.2.2 RL Algorithms

Below, we describe the comparisons from the MDP experiments and their hyperparameters.

**Proximal Policy Optimization (PPO).** The reported results for PPO use the Stable Baselines3 implementation [88] with the default hyperparameters, which successfully learns each task given 100K environment steps in Dark Room and 125K environment steps in Miniworld. In Dark Room, the policy is implemented as a multi-layer perceptron with two hidden layers of 64 units each. In Miniworld, the policy is a convolutional neural network with two convolutional layers with $16\ 3 \times 3$ kernels each, followed by a linear layer with output dimension of 8.

**Algorithm Distillation (AD).** We first collect learning histories with PPO for each of the training tasks. Then, given a cross-episodic context of length $H$, where $H$ is the task horizon, the model is trained to predict the actions taken $K$ episodes later (given the states visited in that episode). This was shown to lead to faster algorithms in [7]. We evaluated AD across different values of $K$. Between $K = 10, 50, 100$, we found $K = 100$ to be most performant in the Dark Room environment. In Miniworld, we also subsampled with $K = 100$. In Dark Room, the transformer has similar hyperparameters as DPT: an embedding size of 32, context length of 100 steps, 4 hidden layers, and 4 attention heads per attention layer. In Miniworld, as with DPT, we first encode the image with a convolutional network with two convolutional layers with $16\ 3 \times 3$ kernels each, followed by a linear layer with output dimension of 8.

**RL$^2$.** The reported results for RL$^2$ use an open-sourced implementation from [22]. The implementation uses PPO as the RL algorithm and defines a single trial as four consecutive episodes. The policy is implemented with one hidden layer of 32 units in Dark Room. In Miniworld, the policy is parameterized with a convolutional neural network with two convolutional layers with $16\ 3 \times 3$ kernels each, followed by a linear layer with output dimension of 8.

**DPT.** The transformer for DPT has an embedding size of 32, context length of 100 steps, 4 hidden layers, and 4 attention heads per attention layer in Dark Room. In Miniworld, the image is first passed through a convolutional network with two convolutional layers $16\ 3 \times 3$ kernels each, followed by a linear layer with output dimension of 8. The transformer model that processes these image embeddings otherwise has the same hyperparameters as in Dark Room. We use the AdamW optimizer with weight decay 1e-4, learning rate 1e-3, and batch-size 128.

### A.3 Bandit Pretraining and Testing

**Basic Bandit.** Offline, to generate the in-context datasets for pretraining, we used a Dirichlet distribution to sample action frequencies in order to generate datasets with diverse compositions (i.e. some more uniform, some that only choose a few actions, etc.): $p_1 \sim \text{Dir}(\mathbb{1})$ where $p_1 \in \Delta(\mathcal{A})$ and $\mathbb{1} \in \mathbb{R}^{|\mathcal{A}|}$. We also mixed this with a distribution that has all mass on one action: $\hat{a} \sim \text{Unif}(\mathcal{A})$ and $p_2(\hat{a}) = 1$ and $p_2(a) = 0$ for all $a \neq \hat{a}$. The final action distribution is $p = (1 - \omega)p_1 + \omega p_2$ where $\omega \sim \text{Unif}(0.1[10])$. We train on 100,000 pretraining samples for 300 epochs with an 80/20 train/validation split. In Figure 2a, $\mathcal{D}_{\text{test}}$ is generated in the same way.

**Expert-Biased Bandit.** To generate expert-biased datasets for pretraining, we compute the action frequencies to bias the dataset towards the optimal action. Let $a^\star$ be the optimal one. As before, we take $p_1 \sim \text{Dir}(\mathbb{1})$. Then, $p_2(a^\star) = 1$ and $p_2(a) = 0$ for all $a \neq a^\star$. For of bias of $\omega$, we take $p = (1 - \omega)p_1 + \omega p_2$ with $\omega \sim \text{Unif}(0.1[10])$. We use the same pretraining sample size and epochs as before. For testing, $\mathcal{D}_{\text{test}}$ is generated the same way except we fix a particular $\omega \in \{0, 0.5, 1\}$ to test on.

**Linear Bandit.** We consider the case where $|\mathcal{A}| = 10$ and $d = 2$. To generate environments from $\mathcal{T}_{\text{pre}}$, we first sampled a fixed set of actions from $\mathcal{N}(\mathbf{0}, I_d/d)$ in $\mathbb{R}^d$ to represent the features. Then, for each $\tau$, we sampled $\theta_\tau \sim \mathcal{N}(\mathbf{0}, I_d/d)$ to produce the means $\mu_a = \langle \theta_\tau, \phi(a) \rangle$ for $a \in \mathcal{A}$. To generate the in-context dataset, we ran Gaussian TS (which does not leverage $\phi$) over $n = 200$ steps (see hyperparameters in previous section). Because order matters, we did not shuffle and used $1,000,000$ pretraining samples over 200 epochs with an 80/20 train/validation split. At test time, we set $\mathcal{T}_{\text{test}} = \mathcal{T}_{\text{pre}}$ and $\mathcal{D}_{\text{test}} = \mathcal{D}_{\text{pre}}$. Note that $\phi$ is fixed over all $\tau$, as is standard for a linear bandit.

### A.4 MDP Environment Details

**Dark Room.** The agent must navigate a $10 \times 10$ grid to find the goal within $H = 100$ steps. The agent's observation is its $xy$-position, the allowed actions are left, right, up, down, and stay, and the reward is only $r = 1$ when the agent is at the goal, and $r = 0$ otherwise. At test time, the agent begins at the $(0, 0)$ position. We randomly designate 80 of the 100 grid squares to be goals for the training tasks, and hold out the remaining 20 for evaluation.

**Miniworld.** The agent must navigate to the correct box, which is initially unknown, from $25 \times 25$ RGB image observations. The agent is additionally conditioned on its own direction vector. In each episode, the environment is initialized with four boxes of different colors, one in each corner of the square room. The agent can turn left, turn right, or move forward. The reward is only $r = 1$ when the agent is near the correct box and $r = 0$ otherwise, and each episode is 50 time-steps long. At test time, the agent begins in the middle of the room.

### A.5 MDP Pretraining Datasets

**Dark Room.** In Dark Room, we collect 100K in-context datasets, each of length $H = 100$ steps, with a uniform-random policy. The 100K datasets are evenly collected across the 100 goals. The query states are uniformly sampled from the state space, and the optimal actions are computed as follows: move up/down until the agent is on the same $y$-position as the goal, then move left/right until the agent is on the $x$-position as the goal. Of the 100K collections of datasets, query states, and optimal actions, we use the first 80K (corresponding to the first 80 goals) for training and the remaining 20K for validation.

**Miniworld.** While this task is solved from image-based observations, we also note that there are only four distinct tasks (one for each colored box), and the agent does not need to handle new tasks at test time. Hence, the number of in-context datasets required in pretraining is fewer – we use 60K datasets each of length $H = 50$ steps. So as to reduce computation, the in-context datasets only have only $(s, a, r)$ tuples. The query states, which consist of image and direction are sampled uniformly from the entire state space, i.e., the agent is place uniformly at random in the environment, pointing in a random direction. The optimal actions are computed as follows: turn towards the correct box if the agent is not yet facing it (within $\pm 15$ degrees), otherwise move forward. Of the 60K collections of datasets, query states, and optimal actions, we use 80% for training and the remaining 20% for validation.

## B  Additional Experimental Results

### B.1  Bandits

This section reports additional experimental results in bandit environments.

**Out-of-distribution reward variances.** In Figures 2c and 6a, we demonstrate the robustness of the basic pretrained model under shifts in the reward distribution at test time by varying the amount of noise observed in the rewards. DPT maintains robustness to these shifts similar to TS.

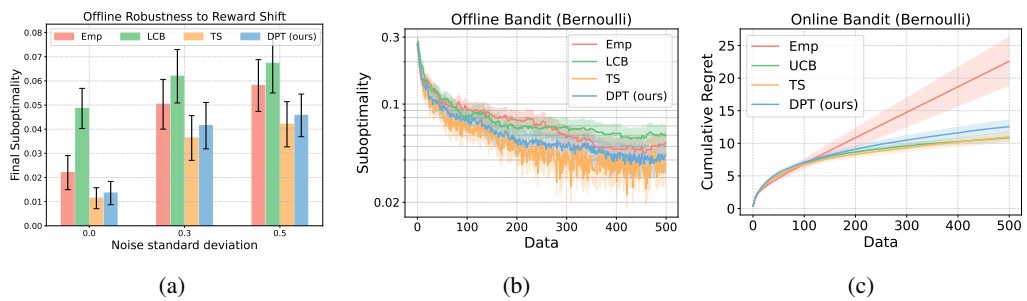

(a)                (b)                (c)

Figure 6: (a) Final (after 500 steps) offline suboptimality on out-of-distribution bandits with different Gaussian noise standard deviations. (b) Offline performance on out-of-distribution Bernoulli bandits, given random in-context datasets. (c) Online cumulative regret on Bernoulli bandits. The mean and standard error are computed over 200 test tasks.

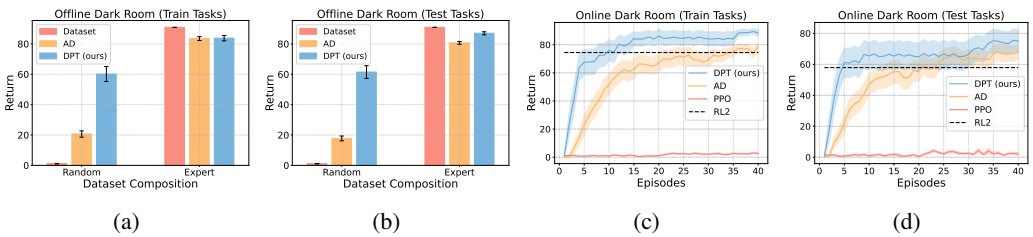

(a)                (b)                (c)                (d)

Figure 7: All comparisons in Dark Room evaluated on the tasks that were seen during pretraining, displayed next to their evaluations on test task counterparts from the main text.

**Bernoulli rewards.**    We test the out-of-distribution ability of DPT further by completely changing the reward distribution from Gaussian to Bernoulli bandits. Despite being trained only on Gaussian tasks during pretraining, DPT maintains strong performance both offline and online in Figures 6b and 6c.

## B.2   Markov Decision Processes

This section reports additional experimental results in the Dark Room and Miniworld environments.

**Performance on training tasks.**    In Fig. 7, we show the performance of each method on the training tasks in Dark Room. Offline, DPT and AD demonstrate comparable performance as on the training tasks, indicating a minimal generalization gap to new goals. Online, DPT, AD, and RL$^2$ also achieve performance on the training tasks similar to that on the test tasks.

**Generalization to new dynamics.**    In this experiment, we study generalization to variations in a different aspect of the MDP, namely the dynamics. We design *Dark Room (Permuted)*, a variant of Dark Room in which the goal is fixed to a corner but the action space is randomly permuted. Hence, the agent must leverage its historical context to infer the effect of each action. On a held-out set of 20 permutations, DPT infers the optimal policy correctly every time offline, given only 100 offline samples, matching the optimal policy at 83 return. Similarly, the online performance immediately snaps to a near optimal policy in one episode once it identifies the novel permutation in Figure 8.

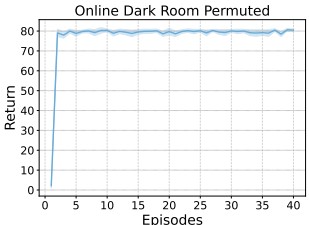

Figure 8: Online evaluation of DPT on Dark Room when tested on novel actions set permutations.

## B.3   Sensitivity Analysis

We next seek to understand the sensitivity of DPT to different hyperparameters, including the model size and size of the pretraining dataset. These experiments are performed in the Dark Room environment. As shown in Fig. 9, the performance of DPT is robust to the model size; it is the same across different embedding sizes,

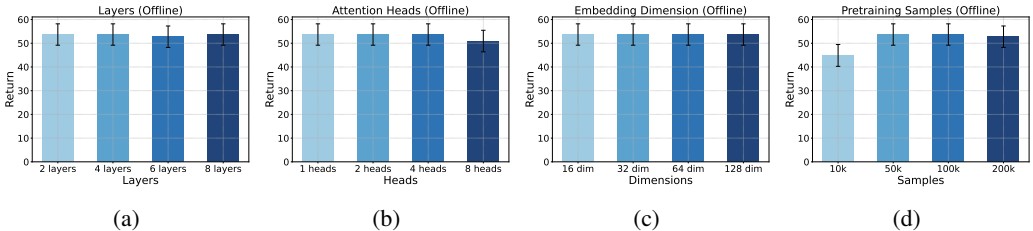

Figure 9: Sensitivity analysis of the offline Dark Rook task over the GPT-2 transformer's hyperparameters: (a) layers (b) attention heads (c) embedding dimensions (d) pretraining samples.

number of layers, and number of attention heads. Notably, the performance is slightly worse with $8$ attention heads, which may be attributed to slight overfitting. We do see that when the pretraining dataset is reduced to $10\%$ of its original size ($10000$ samples) the performance degrades, but otherwise has similar performance with larger pretraining datasets.

## C   Additional Theory and Omitted Proofs

We start with a well-known concentration inequality for the maximum-likelihood estimate (MLE) to provide some more justification for the approximation made in Assumption 1. We state a version from [89]. Let $\mathcal{F}$ be a finite function class used to model a conditional distribution $p_{Y|X}(y|x)$ for $x \in \mathcal{X}$ and $y \in \mathcal{Y}$. Assume there is $f^\star \in \mathcal{F}$ such that $p(y|x) = f^\star(y|x)$ (realizable), and $f(\cdot|x) \in \Delta(\mathcal{Y})$ for all $x \in \mathcal{X}$ and $f \in \mathcal{F}$ (proper). Let $D = \{x_i, y_i\}_{i\in[N]}$ denote a dataset of i.i.d samples where $x_i \sim p_X$ and $y_i \sim p_{Y|X}(\cdot|x_i)$. Let

$$\hat{f} = \underset{f \in \mathcal{F}}{\arg\max} \sum_{i \in [N]} \log f(y_i|x_i) \tag{6}$$

**Proposition C.1** (Theorem 21 of [89])**.** *Let $D$ and $\hat{f}$ be given as above under the aforementioned conditions. Then, with probability at least $1 - \delta$,*

$$\mathbb{E}_{x \sim p_X} \|\hat{f}(\cdot|x) - p_{Y|X}(\cdot|x)\|_1^2 \leq \frac{8 \log\left(|\mathcal{F}|/\delta\right)}{N} \tag{7}$$

The finiteness of $\mathcal{F}$ is done for simplicity, but we can see that this yields dependence on the log-cardinality, a common measure of complexity. Extensions to infinite $\mathcal{F}$ of bounded statistical complexity can be readily made to replace this. For our setting, the bound suggests that $\mathbb{E}_{P_{pre}} \|P_{pre}(\cdot|s_{\text{query}}, D, \xi_h) - M_\theta(\cdot|s_{\text{query}}, D, \xi_h)\|_1^2 \to 0$ as $N \to \infty$ with high probability, provided the function class of $M_\theta$ has bounded statistical complexity.

### C.1   Posterior Sampling

Posterior sampling is most generally described with the following procedure [13]. Initialize a prior distribution $\mathcal{T}_1 = \mathcal{T}_{\text{pre}}$ and dataset $D = \{\}$. For $k \in [K]$

1. Sample $\tau_k \sim \mathcal{T}_k$ and compute $\hat{\pi}_{\tau_k}$
2. Execute $\pi^\star_{\tau_k}$ and add interactions to $D$
3. Update posterior distribution $\mathcal{T}_{k+1}(\tau) = P(\tau|D)$.

The prior and posteriors are typically over models such as reward functions in bandits or transition dynamics in MDPs.

### C.2   Proof of Theorem 1

**Theorem 1** (DPT $\iff$ PS)**.** *Let the above assumptions hold. Then, $P_{ps}(\xi_H \mid D, \tau_c) = P_{M_\theta}(\xi_H \mid D, \tau_c)$ for all trajectories $\xi_H$.*

*Proof.* Without loss of generality, for a task $\tau$, we take $\pi_\tau^\star(\cdot|s)$ to be deterministic and denote the optimal action in state $s$ as $\pi_\tau^\star(s)$. Recall that we consider a fixed current task $\tau_c$ and a fixed in-context dataset $D$. Define $\xi_h = (s_1, a_1, \ldots, s_h, a_h)$.

We now formally state the variant of the full joint distribution from which we sample during pretraining. Let $\tau$ and $D'$ be an arbitrary task and dataset and let $a^\star \in \mathcal{A}$, $s_{\text{query}} \in \mathcal{S}$, $\xi_{H-1} \in (\mathcal{S} \times \mathcal{A})^{H-1}$, and $h \in [0, H-1]$ be arbitrary.

$$P_{pre}(\tau, a^\star, s_{\text{query}}, D', \xi_{H-1}, h) = \mathcal{T}_{\text{pre}}(\tau)\mathcal{D}_{\text{pre}}(D'; \tau)\mathcal{D}_{\text{query}}(s_{\text{query}})\mathfrak{S}_H(s_{1:H})\pi_\tau^\star(a^\star|s_{\text{query}}) \quad (8)$$

$$\times \text{Unif}[0, H-1] \prod_{i \in [H]} \pi_\tau^\star(a_i|s_i) \quad (9)$$

The $\text{Unif}[0, H-1]$ is due to the fact that we sample $h \sim \text{Unif}[0, H-1]$ and then truncate $\xi_h$ from $\xi_{H-1}$ (or, equivalently, sample $\xi_h \sim \mathfrak{S}_h$ directly), marginalizing out the other variables. For $h' \leq h-1$, recall that we also use the notation $\mathfrak{S}_{h'}(s_{1:h'})$ to denote the marginalization of the full joint $\mathfrak{S}_H$. We will eventually work with the posterior of this distribution given the data $D$ and history $\xi_h$:

$$P_{pre}(\tau|D, \xi_h) \propto \mathcal{T}_{\text{pre}}(\tau)\mathcal{D}_{\text{pre}}(D; \tau) \prod_{i \in [h]} \pi_\tau^\star(a_i|s_i) \quad (10)$$

$$\propto P_{pre}(\tau|D) \prod_{i \in [h]} \pi_\tau^\star(a_i|s_i) \quad (11)$$

We define the following random sequences and subsequences:

$$\Xi_{ps}(h; D) = (S_1^{ps}, A_1^{ps}, \ldots, S_h^{ps}, A_h^{ps}) \quad (12)$$

where the variables are generated according to the following conditional process: $\tau_{ps} \sim P(\cdot|D)$, $S_1^{ps} \sim \rho_{\tau_c}$, $A_h^{ps} \sim \pi_{\tau_{ps}}^\star(\cdot|S_h^{ps})$, and $S_{h+1}^{ps} \sim T_{\tau_c}(\cdot|S_h^{ps}, A_h^{ps})$. We also define $\Xi_{ps}(h' : h; D)$ to be the last $h - h'$ elements of $\Xi_{ps}(h; D)$. Analogously, we define

$$\Xi_{pre}(h; D) = (S_1^{pre}, A_1^{pre}, \ldots, S_h^{pre}, A_h^{pre}) \quad (13)$$

where the variables are from the process: $S_1^{pre} \sim \rho_{\tau_c}$, $A_h^{pre} \sim P_{pre}(\cdot|S_h^{pre}, D, \Xi_{pre}(h-1; D))$, and $S_{h+1}^{pre} \sim T_{\tau_c}(\cdot|S_h^{pre}, A_h^{pre})$. Note that $A_h^{pre}$ is sampled conditioned on the sequence $\Xi_{pre}(h; D)$ so far.

We will show that $\Xi_{ps}(h; D)$ and $\Xi_{pre}(h; D)$ follow the same distribution for all $h \in [H]$. For convenience, we will drop notational dependence on $D$, except where it resolves ambiguity. Also, because of Assumption 1, we have that $P_{pre}(\cdot|S_h^{pre}, D, \Xi_{pre}(h-1)) = M_\theta(\cdot|S_h^{pre}, D, \Xi_{pre}(h-1))$, so we will just work with $P_{pre}$ for the remainder of the proof. We will also make use of the following lemma.

**Lemma C.2.** *If $\mathcal{D}_{pre}$ is complaint, then $P_{pre}(\tau|D) = P(\tau_{ps} = \tau|D)$.*

*Proof.* From the definition of posterior sampling (using the same prior, $\mathcal{T}_{\text{pre}}$), we have that

$$P(\tau_{ps} = \tau|D) \propto P(D|\tau)\mathcal{T}_{\text{pre}}(\tau) \quad (14)$$

$$\propto \mathcal{T}_{\text{pre}}(\tau) \prod_{j \in [n]} T_\tau(s_j'|s_j, a_j)R_\tau(r_j|s_j, a_j) \quad (15)$$

$$\propto \mathcal{T}_{\text{pre}}(\tau) \prod_{j \in [n]} T_\tau(s_j'|s_j, a_j)R_\tau(r_j|s_j, a_j)\mathcal{D}_{\text{pre}}(a_j|s_j, D_{j-1}) \quad (16)$$

$$= \mathcal{T}_{\text{pre}}(\tau)\mathcal{D}_{\text{pre}}(D; \tau) \quad (17)$$

$$= P_{pre}(\tau|D) \quad (18)$$

where the second line crucially uses the fact that posterior sampling chooses actions based only on the prior and history so far. Similarly, the third line uses the fact that $\mathcal{D}_{\text{pre}}$ is compliant. Since the two sides are proportional in $\tau$, they are equivalent. □

We will prove Theorem 1 via induction for each $h \in [H]$. First, consider the base case for a sequence of length $h = 1$. Recall that $\rho_{\tau_c}$ denotes the initial state distribution of $\tau_c$. We have that the densities can be written as

$$P(\Xi_{ps}(1) = \xi_1) = P(S_1^{ps} = s_1, A_1^{ps} = a_1) \tag{19}$$

$$= \rho_{\tau_c}(s_1)P(A_1^{ps} = a_1|S_1^{ps} = s_1) \tag{20}$$

$$= \rho_{\tau_c}(s_1)\int_\tau P(A_1^{ps} = a_1, \tau_{ps} = \tau|S_1^{ps} = s_1)d\tau \tag{21}$$

$$= \rho_{\tau_c}(s_1)\int_\tau \pi_\tau^\star(a_1|s_1)P_{ps}(\tau_{ps} = \tau|D, S_1^{ps} = s_1)d\tau \tag{22}$$

$$= \rho_{\tau_c}(s_1)\int_\tau \pi_\tau^\star(a_1|s_1)P_{ps}(\tau_{ps} = \tau|D)d\tau \tag{23}$$

$$= \rho_{\tau_c}(s_1)P_{pre}(A_1^{pre} = a_1|s_1, D) \tag{24}$$

$$= P(\Xi_{pre}(1) = \xi_1) \tag{25}$$

where the second line uses the sampling process of $S_1^{pre}$; the third marginalizes over $\tau_{ps}$, which is the task that posterior sampling samples to find the optimal policy; the fourth decomposes this into the optimal policy and the posterior over $\tau_{ps}$ given $D$ and $S_1^{ps}$. Since $S_1^{ps}$ is independent of sampling of $\tau_{ps}$ this dependence goes away in the next line. The sixth line applies Lemma C.2 and then, for $h = 1$, there is no history to condition on.

Now, we leverage the inductive hypothesis to prove the full statement. Suppose that the hypothesis holds for $h - 1$. Then,

$$P(\Xi_{ps}(h) = \xi_h) = P(\Xi_{ps}(h-1) = \xi_{h-1})P(S_h^{ps} = s_h, A_h^{ps} = a_h|\Xi_{ps}(h-1) = \xi_{h-1}) \tag{26}$$
$$\tag{27}$$

By the hypothesis, we have that $P(\Xi_{ps}(h-1) = \xi_{h-1}) = P(\Xi_{pre}(h-1) = \xi_{h-1})$. For the second factor,

$$P(S_h^{ps} = s_h, A_h^{ps} = a_h|\Xi_{ps}(h-1) = \xi_{h-1}) \tag{28}$$

$$= T_{\tau_c}(s_h|s_{h-1}, a_{h-1}) \cdot P(A_h^{ps} = a_h|S_h^{ps} = s_h, \Xi_{ps}(h-1) = \xi_{h-1}) \tag{29}$$

$$= T_{\tau_c}(s_h|s_{h-1}, a_{h-1}) \cdot \int_\tau P(A_h^{ps} = a_h, \tau_{ps} = \tau|S_h^{ps} = s_h, \Xi_{ps}(h-1) = \xi_{h-1})d\tau \tag{30}$$

As before, we can further rewrite the last factor as

$$P(A_h^{ps} = a_h, \tau_{ps} = \tau|S_h^{ps} = s_h, \Xi_{ps}(h-1) = \xi_{h-1}) \tag{31}$$

$$= \pi_\tau^\star(a_h|s_h) \cdot P(\tau_{ps} = \tau|S_h^{ps} = s_h, \Xi_{ps}(h-1) = \xi_{h-1}) \tag{32}$$

where

$$P(\tau_{ps} = \tau|S_h^{ps} = s_h, \Xi_{ps}(h-1) = \xi_{h-1}) = \frac{P(S_h^{ps} = s_h, \Xi_{ps}(h-1) = \xi_{h-1}|\tau_{ps} = \tau)P(\tau_{ps} = \tau|D)}{P(S_h^{ps} = s_h, \Xi_{ps}(h-1) = \xi_{h-1})}$$
$$\tag{33}$$

$$\propto P_{pre}(\tau|D) \prod_{i\in[h-1]} T_{\tau_c}(s_{i+1}|s_i, a_i)\pi_\tau^\star(a_i|s_i) \tag{34}$$

$$\propto P_{pre}(\tau|D) \prod_{i\in[h-1]} \pi_\tau^\star(a_i|s_i) \tag{35}$$

$$\propto P_{pre}(\tau|D)\mathcal{D}_{\text{query}}(s_h)\mathfrak{S}_{h-1}(s_{1:h-1}) \prod_{i\in[h-1]} \pi_\tau^\star(a_i|s_i) \tag{36}$$

$$\propto P_{pre}(\tau|s_h, D, \xi_{h-1}) \tag{37}$$
$$\tag{38}$$

where $\propto$ denotes that the two sides are equal up to multiplicative factors independent of $\tau$. In the first line, we used Bayes rule. In the second line, given that $\tau_{ps} = \tau$ (i.e. posterior sampling selected $\tau$ to

deploy), we decompose the probability of observing that sequence of states of actions. We also used Lemma C.2. The denominator does not depend on $\tau$. Similarly, for the third and fourth lines, $T_{\tau_c}$ and $\mathfrak{S}$ do not depend on $\tau$. The final line follows from the definition of the joint pretraining distribution in this regime.

Therefore, we conclude that the posterior over the value of $\tau_{ps}$ is the same as the posterior over the task in the pretraining distribution, given $s_h, D, \xi_{h-1}$. Substituting back through all the previous equations, we have

$$P(\Xi_{ps}(h) = \xi_h) \tag{39}$$

$$= P(\Xi_{pre}(h-1) = \xi_{h-1}) \cdot T_{\tau_c}(s_h|s_{h-1}, a_{h-1}) \int_\tau \pi_\tau^\star(a_h|s_h) P_{pre}(\tau|s_h, D, \xi_{h-1}) d\tau \tag{40}$$

$$= P(\Xi_{pre}(h-1) = \xi_{h-1}) \cdot T_{\tau_c}(s_h|s_{h-1}, a_{h-1}) P_{pre}(a_h|s_h, D, \xi_{h-1}) \tag{41}$$

$$= P(\Xi_{pre}(h) = \xi_h) \tag{42}$$

This concludes the proof.

$\square$

## C.3   Proof of Corollary 6.2

**Corollary C.3** (Finite MDPs). *Suppose that $\sup_\tau \mathcal{T}_{test}(\tau)/\mathcal{T}_{pre}(\tau) \leq \mathcal{C}$ for some $\mathcal{C} > 0$. For the above MDP setting, the pretrained model $M_\theta$ satisfies $\mathbb{E}_{\mathcal{T}_{test}}[Reg_\tau(M_\theta)] \leq \widetilde{\mathcal{O}}(\mathcal{C}H^{3/2}S\sqrt{AK})$.*

*Proof.* Note that $\mathcal{D}_{\mathrm{pre}}$ is clearly compliant since it is generated by random sampling. We use the equivalence between $M_\theta$ and posterior sampling established in Theorem 1. The proof then follows immediately from Theorem 1 of [13] to guarantee that

$$\mathbb{E}_{\mathcal{T}_{\mathrm{pre}}}[\mathrm{Reg}_\tau(M_\theta)] \leq \widetilde{\mathcal{O}}\left(H^{3/2}S\sqrt{AK}\right) \tag{43}$$

where the notation $\widetilde{\mathcal{O}}$ omits polylogarithmic dependence. The bound on the test task distribution follows from the assumed bound on the likelihood ratio under the priors:

$$\int \mathcal{T}_{\mathrm{test}}(\tau)\mathrm{Reg}_\tau(M_\theta)d\tau \leq \mathcal{C}\int \mathcal{T}_{\mathrm{pre}}(\tau)\mathrm{Reg}_\tau(M_\theta)d\tau. \tag{44}$$

$\square$

## C.4   Proof of Corollary 6.3

**Corollary C.4** (Latent representation learning in linear bandits). *For $\mathcal{T}_{test} = \mathcal{T}_{pre}$ in the above linear bandit setting, $M_\theta$ satisfies $\mathbb{E}_{\mathcal{T}_{test}}[Reg_\tau(M_\theta)] \leq \widetilde{\mathcal{O}}(d\sqrt{K})$.*

*Proof.* The distribution $\mathcal{D}_{\mathrm{pre}}$ satisfies compliance by definition because it is generated by an adaptive algorithm TS. The proof once again follows by immediately deferring to the established result of [68] (Proposition 3) for linear bandits by the posterior sampling equivalence of Theorem 1. This ensures that posterior sampling achieves regret $\widetilde{\mathcal{O}}(d\sqrt{K})$. It remains, however, to justify that $P_{pre}(\cdot|D_k)$ will be covered by Gaussian Thompson Sampling for all $D_k$ with $k \in [K]$. This is verified by noting that $P_{ps}(a|D_k) > 0$ for non-degenerate Gaussian Thompson Sampling (positive variances of the prior and likelihood functions) and finite $K$. This guarantees that any $D_k$ will have support. $\square$

## C.5   Proof of Proposition 6.4

**Proposition C.5.** *Let $P_{pre}^1$ and $P_{pre}^2$ be pretraining distributions that differ only by their in-context dataset distributions, denoted by $\mathcal{D}_{pre}^1$ and $\mathcal{D}_{pre}^2$. If $\mathcal{D}_{pre}^1$ and $\mathcal{D}_{pre}^2$ are compliant with the same support, then $P_{pre}^1(a^\star|s_{query}, D, \xi_h) = P_{pre}^2(a^\star|s_{query}, D, \xi_h)$ for all $a^\star, s_{query}, D, \xi_h$.*

*Proof.* The proof follows by direct inspection of the pretraining distributions. For $P_{pre}^1$, we have

$$P_{pre}^1(a^\star|s_{\text{query}}, D, \xi) = \int_\tau \pi_\tau^\star(a^\star|s_{\text{query}}) P_{pre}^1(\tau|s_{\text{query}}, D, \xi) d\tau \tag{45}$$

The posterior distribution over tasks is simply

$$P_{pre}^1(\tau|s_{\text{query}}, D, \xi) = \frac{P_{pre}^1(\tau, s_{\text{query}}, D, \xi)}{P_{pre}^1(s_{\text{query}}, D, \xi)} \tag{46}$$

$$\propto P_{pre}^1(\tau) P_{pre}^1(\xi|\tau) \mathcal{D}_{\text{query}}(s_{\text{query}}) \mathcal{D}_{\text{pre}}^1(D; \tau) \tag{47}$$

$$= P_{pre}^2(\tau) P_{pre}^2(\xi|\tau) \mathcal{D}_{\text{query}}(s_{\text{query}}) \mathcal{D}_{\text{pre}}^1(D; \tau) \tag{48}$$

Then, the distribution over the in-context dataset can be decomposed as

$$\mathcal{D}_{pre}^1(D; \tau) = \prod_{i \in [n]} R_\tau(r_i|s_i, a_i) T_\tau(s_i'|s_i, a_i) \mathcal{D}_{\text{pre}}^1(a_i|s_i, D_{i-1}; \tau) \tag{49}$$

$$= \prod_{i \in [n]} R_\tau(r_i|s_i, a_i) T_\tau(s_i'|s_i, a_i) \mathcal{D}_{\text{pre}}^1(a_i|s_i, D_{i-1}) \tag{50}$$

$$\propto \prod_{i \in [n]} R_\tau(r_i|s_i, a_i) T_\tau(s_i'|s_i, a_i) \mathcal{D}_{\text{pre}}^2(a_i|s_i, D_{i-1}) \tag{51}$$

$$= \mathcal{D}_{pre}^2(D; \tau), \tag{52}$$

where the second equality holds because $\mathcal{D}_{\text{pre}}^1(a_j|s_j, D_j; \tau)$ is assumed to be invariant to $\tau$ by compliance, and the fifth equality holds because $\mathcal{D}_{\text{pre}}^2(a_j|s_j, D_j; \tau)$ is assumed to be invariant to $\tau$.

Therefore, we conclude that, for any $s, D, \xi$,

$$P_{pre}^1(\tau|s, D, \xi) \propto P_{pre}^2(\tau) P_{pre}^2(\xi|\tau) \mathcal{D}_{\text{query}}(s) \mathcal{D}_{\text{pre}}^2(D; \tau) \tag{53}$$

$$\propto P_{pre}^2(\tau|s, D, \xi). \tag{54}$$

Since also $\int_\tau P_{pre}^1(\tau|s, D, \xi) = 1 = \int_\tau P_{pre}^2(\tau|s, D, \xi)$, then

$$P_{pre}^1(\tau|s, D, \xi) = P_{pre}^2(\tau|s, D, \xi). \tag{55}$$

Substituting this back into Equation 45 yields $P_{pre}^1(a^\star|s, D, \xi) = P_{pre}^1(a^\star|s, D, \xi)$. □

