# OpenReview forum: "Supervised Pretraining Can Learn In-Context Reinforcement Learning"
_NeurIPS.cc/2023/Conference — NeurIPS 2023 spotlight_

### Official Review · Reviewer_qnXx · 2023-07-03

**Soundness:** 3 good
**Presentation:** 3 good
**Contribution:** 3 good
**Rating:** 6
**Confidence:** 3

**Summary:**

This paper introduces a new pretraining objective for decision-making tasks. Leveraging the in-context learning capabilities of transformers, the authors propose to pretrain a model that predicts the optimal action given the current state and a set of history interactions as context. Empirically, the pretrained model is capable of learning various algorithms implicitly and can transfer to unseen tasks. Theoretically, the pretraining objective can be viewed as training the model to perform posterior sampling.

**After rebuttal**: The rebuttal addressed my concerns sufficiently. The new experimental results highlight the importance of both in-context datasets and optimal actions. I have increased my score to 6.

**Strengths:**

1. This work proposes a new pretraining framework for decision-making tasks. It is one of the pioneering efforts in the problem of in-context decision-making.
2. The paper is generally sound and presents adequate empirical results. The connection to posterior sampling is also interesting.
3. The paper is well-written.

**Weaknesses:**

1. My main concern is about the requirement of optimal actions at pretraining. This assumption appears to be overly strong and I doubt its scalability to large-scale pretraining and more complex tasks.
2. I find it unclear whether the pretrained model benefits more from in-context datasets or optimal actions. It is suggested to conduct an ablation study on the context size, which appears to be missing from the appendix. I am curious about the performance when the context size is set to 0.
3. The modification of history-dependent pretraining in Section 6 needs further clarification. The importance of this modification is unclear to me.
4. The authors should provide a more comprehensive comparison between their proposed pretraining objective and previous research, such as retrieval-augmented RL [1].

[1] Goyal A, Friesen A, Banino A, et al. Retrieval-augmented reinforcement learning[C]//International Conference on Machine Learning. PMLR, 2022: 7740-7765.

**Questions:**

1. In Appendix A.5, you mention that the in-context datasets for Miniworld do not include images as observations. Does this imply that in-context datasets are not important as expected for the final performance?
2. Can you please explain the difference between DPT(PPO, PPO) and AD, and why DPT(PPO, PPO) is superior to AD?
3. How are the datasets constructed for Dark Room (Three Tasks)?

**Limitations:**

The authors acknowledge certain limitations of the proposed method.

---

> ### Author Rebuttal · Authors · 2023-08-09
>
> Thanks for detailed feedback and questions!
>
> **Optimal actions at pretraining.**: Indeed, the theoretical formulation of DPT requires that task-specific optimal actions can be obtained for pretraining. While restrictive in some cases, we actually view this as being potentially quite flexible. This is because these labels can be obtained *offline* through many different means:
> - If task-specific expert labelers or demonstrations are available, those can be used directly
> - If the data is collected via a (inefficient or efficient) single-task RL algorithm, we can retroactively use labels from the learned policy (Fig 5.bc, also Fig 3.bc, Cor 6.2)
> - If privileged information is only available during pretraining but not testing (e.g. synthetic tasks, sim-to-real, hindsight [1]), we can obtain action labels that would otherwise be too hard to obtain
> - If the data covers the optimal policy distribution, offline RL can compute a near-optimal policy (same setting as [2]).
>
> In all cases, one can “relabel” the data with the optimal action in a query state. A key feature of DPT is that it is agnostic to how the training tasks are solved as long as the labels are found. The theory of DPT does not require demonstrations from an optimal RL algorithm, only task-specific optimal actions.
>
> Many meta-RL strategies also implicitly assume the training tasks can be “solved,” but they often require that they are solved in a specific way (e.g. online like PEARL [3] or by using a realizable source algorithm like AD).
>
> Perhaps more importantly, DPT can work with approximately optimal actions. The procedure is agnostic to whether the provided labels are actually optimal or not. We showed empirically that if we only know “good” actions such as those from a final PPO policy, DPT still can work well (see Figs 5.bc).
>
> [1] Sinclair et al. Hindsight learning for mdps with exogenous inputs. 2023.
>
> [2] Mendonca et al. Guided meta-policy search. 2019.
>
> [3] Rakelly et al. Efficient off-policy meta-RL via probabilistic context variables. 2019.
>
> **Does pretraining benefit more from in-context datasets or optimal actions? What if the context size is set to 0?**: Both are critical. The in-context dataset provides information about the task through rewards and transitions. It serves the same purpose as an offline dataset for an offline RL or a history for online RL. With no data in a new task, DPT makes a guess from its learned prior. Information-theoretically, no algorithm can do better than this. The prediction becomes more refined with more data or online interactions. See Fig 2.a at $n = 0$ for offline bandits and Fig 2.b for online. For MDPs, see Figs 4.bd. We also included a new figure for offline Dark Room with DPT in the response PDF (Fig 2.c).
>
> Optimal actions are important to connect with posterior sampling (PS) on new RL problems. Again, approximate ones suffice in practice. Pretraining to predict actions only in the in-context datasets would be equivalent to AD.
>
>
> **The modification of history-dependent pretraining in Section 6 needs further clarification.**: Thanks for pointing this out. The context is augmented with a history of optimal actions and states up to the current timestep during pretraining. During deployment, $D$ behaves the same, and the learner’s in-episode history is recorded in $\xi$. This modification is important to establish the precise connection between DPT and PS. In particular, conditioning on the history allows us to ‘collapse’ the posterior distribution over actions that are only consistent with policies that could have been generated by PS.
>
>
> **Comparison to retrieval-augmented RL [1]**: Thank you for pointing out this relevant work. [1] augments a standard RL algorithm with a dataset retrieval process to supply additional immediate information for the agent to use. They demonstrate this can also be used for multitask settings. DPT learns directly to map datasets and states to actions via offline supervised pretraining, and we also provide a theoretical analysis.
>
> **The in-context datasets for Miniworld do not include images as observations. Does this imply that in-context datasets are not important as expected for the final performance?**: We apologize for the confusion. To make this experiment more clear, we have included a new version in the response PDF (Fig 2.ab) which uses images and completely removes the xy-position information. To answer your question, in the original version, the in-context datasets provide contextual information about the task through the xy-position + rewards. Observations are indeed necessary to identify and solve the task.
>
> **Difference between DPT(PPO, PPO) and AD, and why DPT(PPO, PPO) is superior to AD?**: Both use data from PPO. A key difference is that AD is directly trained on the trajectories generated by PPO (i.e. to predict the next action from the learning algorithm), but DPT(PPO, PPO) is trained to predict the action from the *final PPO policy*, given an offline history and a state. This action acts as an approximation of the optimal policy.  AD will only be as good as the algorithm it imitates, but DPT may be better because it can act as a good approximation of posterior sampling with a data-driven prior. [Note that the AD authors consider skipping trajectories to predict slightly further ahead, which we implemented.]
>
>
> **How are the datasets constructed for Dark Room (Three Tasks)?**: The three tasks in Dark Room (Three Tasks) differ in the start and goal location of the agent: (1) start at (0, 5) and goal at (9, 5); (2) start at (5, 0) and goal at (5, 9); (3) start at (0, 5) and goal at (5, 9). The pretraining in-context dataset here is collected by a random policy for the two goals. The in-context dataset consists of demonstrations of the first two tasks, where the reward is 1 only for transitions to (5, 9), and our evaluation is in the third task. This demonstrates the ability to do offline stitching of suboptimal demonstrations.

---

> > ### Comment · Reviewer_qnXx · 2023-08-16
> >
> > Thank you for your comprehensive response to my concerns. Given the convincing experimental results highlighting the importance of both in-context datasets and optimal actions, I am inclined to increase my score to 6.

---

### Official Review · Reviewer_R3a7 · 2023-07-05

**Soundness:** 3 good
**Presentation:** 4 excellent
**Contribution:** 3 good
**Rating:** 8
**Confidence:** 3

**Summary:**

This paper proposes an in-context learning based algorithm for meta-learning decision-making algorithms, combined with a novel analysis which shows that unlike prior works, the proposed algorithm is guaranteed to work. Moreover, the authors also show empirically and theoretically that their algorithm is able to improve upon its training demonstrations.

**Strengths:**

1. The paper conducts a novel theoretical analysis on the ability to generalize well when meta-learning with the in-context learning paradigm applied to decision-making problems.

2. The latent representation learning results in linear bandits which enable the improvement over the training demonstrations is a novel motivation for in-context learning based decision-making algorithms.

3. The paper is clearly written and accessible to readers with varying backgrounds.

**Weaknesses:**

The formulation of a consistent learned model assumes the meta-learning algorithm is perfect. Can you relax this assumption to provide the fine-grained pre-training sample complexity?

**Questions:**

Could you please explain why in Figure 2, UCB is outperformed by Thompson Sampling even though you stated that UCB is an optimal online multi-armed bandit algorithm?

**Limitations:**

Since the analysis for MDP assumes that the training demonstrations are generated with an optimal policy, the claim that the proposed method can theoretically improve upon the training demonstrations should explicitly specify that the improvement is only for the bandit and contextual bandit cases, not the more general MDP case.

---

> ### Author Rebuttal · Authors · 2023-08-09
>
> Thank you for the positive comments and detailed feedback!
>
> **Can you relax the consistency assumption on the learned model for the theory?**: Thank you for the great question. In Appendix C, we provide some justification for this via a generic MLE concentration inequality (cf. Zhang, 2006). This will be true for realizable function classes of bounded statistical complexity. Asymptotically (with N, the number of pretraining samples), the distribution can be learned through this MLE procedure. While finite sample versions can be derived [2], the calculations are long and can obfuscate the key theoretical takeaways of the theorem and proof, which is primarily about how the pretraining can lead to in-context posterior sampling behavior. While indeed extremely important, the study of the generalizability and complexity of transformers is in early stages and beyond the scope of this paper. This assumption also allows the theory to be somewhat agnostic to the underlying model used to learn the distribution.
>
> [1] Zhang. From ɛ-entropy to KL-entropy: Analysis of minimum information complexity density estimation. 2006.
>
> [2] Du et al. Few-shot learning via learning the representation, provably. 2020.
>
> **Why is UCB outperformed by Thompson Sampling when it is said to be optimal?**: We apologize we were unclear and will update the draft to clarify this statement. UCB is theoretically *minimax* optimal for regret in the sense that its proven regret *rates* match an information-theoretic lower bound, up to log factors and constants. TS is also minimax optimal. In specific instances TS or UCB may outperform each other.
>
>  **Since the analysis for MDP assumes that the training demonstrations are generated with an optimal policy, the claim that the proposed method can theoretically improve upon the training demonstrations should explicitly specify that the improvement is only for the bandit and contextual bandit cases, not the more general MDP case**:
> To clarify, we do not assume the in-context datasets are necessarily generated with an optimal policy or optimal RL algorithm. For example, a random algorithm could be executed in a well-connected tabular MDP and, if there is sufficient data, one can extract the task-specific optimal policy from collected data. In such settings, DPT can learn a RL strategy that is better than the random strategy used to generate data for the pretraining process, so that it can solve new RL problems more efficiently than the random algorithm. When we refer to it improving over pretraining data, we mean it can learn an RL strategy that may be better than the process used to generate the pretraining data (which could be an inefficient RL algorithm). We do not require demonstrations of an optimal RL algorithm. For MDPs, this is evidenced by Corollary 6.1 (pretraining in-context datasets generated by random interactions) and Figs 5.bc (pretraining data generated by PPO). We will be sure to articulate this point better in the revision.

---

> > ### Comment · Reviewer_R3a7 · 2023-08-20
> >
> > Thanks for the thorough rebuttal. Authors have addressed all my concerns and clarified misunderstandings. I am thus willing to increase my rating.

---

### Official Review · Reviewer_pHmJ · 2023-07-07

**Soundness:** 3 good
**Presentation:** 4 excellent
**Contribution:** 3 good
**Rating:** 7
**Confidence:** 4

**Summary:**

This paper studies in-context learning for sequential decision-making problems. It proposes an approach, Decision Pretrained Transformer (DPT), which takes as input a (state, context) pair and produces an action. It is pretrained on a large number of tasks drawn from some task distribution and evaluated on held-out tasks. This approach can be applied in both offline settings (in which case the context is a given set of trajectories from the task at hand) and online settings (in which case the context is the agent’s experience on the current task so far).

The approach is evaluated in two settings: toy bandits (including linear contextual bandits) and simple MDPs. In the online bandit setting, it is shown to learn behaviors which efficiently balance exploration and exploitation, similarly to hand-coded bandit algorithms like UCB/LinUCB. This hold both in the stateless bandit setting and the linear contextual bandit setting. In the offline setting, it also matches Thompson sampling.

The second setting the algorithm is evaluated on is simple MDPs, namely a small partially observed grid world and MiniWorld (a pixel-based version of Minigrid). Here again, DPT is able to perform well both online and offline, even if the dataset quality is poor.

Finally, the paper shows that under certain assumptions, DPT can be shown to perform posterior sampling, which is consistent with the empirical results in the bandit settings.

Overall, this is pretty good paper. The idea, though simple, has not to my knowledge been explored before. The experiments are well-designed, illustrate the claims well and the writing is clear. The main downside is that the tasks feel pretty toy - the bandit examples are just Gaussians and the MDPs are grid worlds (possibly dressed up with pixels, as in MiniWorld). However, as a first proof of concept I think this still meets the bar for acceptance at NeurIPS.

**Strengths:**

- The problem is definitely interesting, since large-scale, diverse datasets are often available for pretraining and zero-shot adaptation to new tasks is often desirable
- The writing is very clear, and the experiments are well presented
- The experiments are well-designed to show the properties of the proposed approach

**Weaknesses:**

- The experiments are on toy environments
- There is no mention of open sourcing the code. I think this is important since the setting is new and future work should be able to compare to this approach on the same tasks.

**Questions:**

- How does the computational cost of the proposed method compare to that of the baselines? Due to the need to perform attention over the context each step, this might be expensive. Please add a short discussion of this.

- Will the code to reproduce experiments be open-sourced?

**Limitations:**

Yes.

---

> ### Author Rebuttal · Authors · 2023-08-09
>
> Thank you for your positive comments and detailed feedback!
>
> **How does the computational cost of the proposed method compare to that of the baselines? Due to the need to perform attention… this might be expensive**: AD also uses a GPT-like architecture, so its computational cost is approximately the same as DPT. RL^2 uses an RNN which will do inference faster (although progress in transformers is rapidly trying to close this gap). PPO is perhaps the fastest with a simple MLP, but of course it does not benefit from pretraining.
>
> Fortunately, driven by the surge of interest in language models, there is considerable effort in the community to develop fast computational methods for inference with either transformers or other architectures. Known examples include FlashAttention and S4. There is already work beginning to address this for in-context RL [Lu et al, 2023]. We expect the general framework of DPT to immediately inherit benefits from these advances.
>
> **Will the code to reproduce experiments be open-sourced**: Yes, we intend to open-source the code to reproduce the experiments in a convenient and easily extensible manner. In the meantime, original code may be found in the supplementary material for the reviewers’ convenience.
>
> **Experiments on toy environments**: The environments are indeed simple, but also we believe thoroughly studying the problem empirically and theoretically with simple models is key to developing generalizable and still non-trivial insights about it. We look forward to scaling these insights in future work!

---

### Official Review · Reviewer_CX5S · 2023-07-09

**Soundness:** 3 good
**Presentation:** 2 fair
**Contribution:** 1 poor
**Rating:** 6
**Confidence:** 3

**Summary:**


This work leverages transformers to solve decision making problems in a few-shot manner. In particular, they train transformers to predict optimal actions from a task given a query state and a few-shot dataset of state, action, reward transitions. The approach is tested in both multi-armed-bandit problems and POMDPs, showing superior performance than other RL and Meta-RL approaches. It also provides theoretical analysis of the proposed policy, providing regret bounds and guarantees of performance when different algorithms are used to generate the pre-training data.

**Strengths:**

Method:
The work conducts a thorough study of the effectiveness of transformer for in-context learning in decision making problems. While the domains chosen are quite simple, there are some valuable learnings (especially if they transfer to more complex domains), such as:
- The value of sampling actions from the transformer's likelihood in multi-armed bandits
- The effect of using expert datasets for in-context learning
- Better generalization than competing baselines, when tested on new tasks

The work also includes examples of POMDPS with higher dimensional observations (25x25 pixels) showing the proposed pretraining paradigm achieves better performance than competing baselines.

Valuable analysis of the performance of in-context learning when having access to PPO-trained policies, bringing the setting closer to AD. I particularly value how authors evaluate the effect of different sources for the context states and actions. I am wondering whether PPO, PPO is directly comparable to AD though since in their case they provide in-context actions and trajectories for different stages of training.

By chosing simple domains, the work can thoroughly explore the behavior of in-context learning for different properties of the training data, particularly in the multi-armed bandit section. To my knowledge, this is the first work to build connections of in-context learning in decision making problems and posterior sampling, which allows to:
1. Provide an upper bound on the regret of the proposed transformer-based policy.
2. Show that if the pretraining in-context dataset comes from policies that were only trained on the data and task present in the dataset, the final policy will be the same.

Clarity:
Generally clear writing and background. The experiments are simple but informative, and the work is easily reproducible. More notation and clarity should be given in the theory section, such as stating what is H, and providing more intuition in the propositions.



**Weaknesses:**

Novelty: while this paper conducts a reasonable study on the advantages of in-context learning in decision making problems, this is a feature that has been recently studied in more complex domains and tasks (see related works next). My main concern in this work is that there is not much novelty or learnings compared to those works, and that despite making a reasonable study of the capabilities of transformers for few-shot decision making problems, they are studied in very simple scenarios.

Related work: Several works have studied in context-learning for decision making problems, in more complex scenarios [1, 2]. These works should be cited. I am also not clear on what novelty this work provides with respect to the aforementioned papers. I would like a clarification on this as well.

[1] https://arxiv.org/pdf/2206.13499.pdf
[2] https://arxiv.org/abs/2301.07608

Baselines: While RL2 is a fair baseline, I think authors should look at other meta-RL algorithms, or tune the existing baselines to be more comparable. I understand that PPO is used as a reference, but it should at least be finetuned with the few-shot examples given - it is hard to compare methods when some are actually seeing less data. AD is designed to learn to do RL and therefore works best when having a curriculum of trajectories, which is not the case here.

**Questions:**

It is not intuitive to me that a model trained with expert data (DPT-Exp) behaves poorly when the % of expert data is high (Figure 3.a) since that is what is seen during training, I would expect it to b heave the best. Could authors elaborate on why this is the case?


The setting in lines 312-317 makes a lot of sense. Why not using that in all the experiments, if as claimed "the original pretraining method can be seen as a simpler approximation
316 of this modified method"?

I don't understand assumption 1.1, if we are following that assumption, why not directly using P_{pre} as out policy? Some intuition would be helpful.

Proposition 6.3 is very unitnuitive to me, if I understand it correctly. Is it stating that a pretraining dataset coming from PPO and one coming from a random policy would perform the same? I don't see anything in the proposition that implies otherwise, but this doesnt match in the empirical experiments.

**Limitations:**

Yes

---

> ### Author Rebuttal · Authors · 2023-08-09
>
> Thank you for your detailed feedback!
>
> **Novelty**: We apologize this was not clear. DPT indeed offers significant novel insights, distinct from prior work.
> 1. It is not necessary to start with a good RL algorithm to learn a good RL algorithm. Supervised pretraining can produce one. Prior work has assumed access to good data through either a good online RL algorithm or expert demos at test-time. Instead we make the much more flexible assumption that one can compute task-specific (approximately) optimal actions for the histories *offline during pretraining*. This can be done under different assumptions, subsuming prior work, including:
>    - data collected from a (inefficient or efficient) single-task RL algorithm (Figs 5.bc, 3.bc, Cor 6.2). The final learned policy can be used to relabel the training trajectories (Figs 5.bc)
>    - data gathered in a way that includes the task-specific optimal policy distribution, which can then be used to compute the optimal policy via offline RL
>    - privileged information (e.g. sim-to-real, hindsight info) or task-specific expert labelers
> 2. DPT shows provably sample-efficient posterior sampling (an algorithm that has long been studied primarily theoretically) can be scaled up via transformers and *supervised* pretraining (Thm 1, Cor 6.1, Figs 2.ab)
> 3. DPT offers a new way of training a transformer, showing that learning distributions over task-specific optimal actions (rather than imitating existing algorithms) better leverages data-driven priors (Figs 3.abc, 4.b, 5.bc)
>
> We do not believe these important and sometimes surprising implications have been empirically or theoretically elucidated before, in either simple or complex settings.
>
> **Comparison to [1] and [2]**: Thanks for recommending these papers. In addition to the aforementioned new insights, there are technical differences in operation.
>
> - Prompt-DT [1] is designed for offline RL and conditions on expert demos and return-to-go in the context. This necessitates expert demos as input at test-time. DPT is meant for *both* online and offline in-context RL and does not require test-time demos (but it can benefit from them e.g. Figs 4.ac). DPT does not use return-to-go, which saves a test-time hyperparameter. Pretraining is also different as DPT predicts optimal actions from arbitrary datasets.
> - AdA [2] meta learns online (like RL^2). Major differences are (1) a base online RL alg to optimize and online access to the simulator, and (2) a curriculum learning method. DPT is learned from offline supervised pretraining and it is agnostic to how the pretraining data is acquired. The curriculum is important but orthogonal to our contributions. We believe many of the ideas in AdA are complementary and can be ported to DPT.
>
> **Baselines**: Thanks for your suggestions. We are not aware of an immediate extension of PPO to few-shot finetuning. Training on different tasks does not work because the policy cannot know its task only from state. Using history effectively would make it RL^2. We presume the reviewer is suggesting to do something like MAML. We have included MAML in the response PDF (Fig 1.a). MAML is notably worse and this has been observed before in exploration problems [1]. We inspected the pre-adaptation episodes (Fig 1.b) and found that they fail to explore the task space.
>
> We also included a new comparison with Prompt-DT (Fig 1.c). Prompt-DT requires expert demos and only works offline, whereas DPT works with expert or random data and works online.
>
> [1] Gupta et al. Meta-RL of Structured Exploration Strategies. 2018.
>
> **AD implementation**: We apologize for any confusion. We believe our implementation of AD is already consistent with both your suggestion and the recipe in the original paper. We distill PPO histories by predicting the next actions. As the authors suggest, we apply skipping to improve adaptation.
>
> **DPT-Exp performance in Fig 3.a**: We believe there may be a misunderstanding. Fig 3.a shows that the suboptimality of DPT-Exp goes to zero as the expert data % is increased. Lower is better. This appears to be consistent with what the reviewer has expected.
>
> **Why not do modified DPT (presented in the theory) in the experiments?**: While theoretically elegant, the modified version is not the most practical for a transformer. It is possible to implement, but leaves open some ambiguous design choices. E.g. one should differentiate the in-context dataset D from the history $\xi$ and $\xi$ should have positional encoding. In practice, it is easier to implement the original, which still has strong empirical performance (and is still equivalent to PS for bandits).
>
> **If we are following Asmp 1, why not directly use P_{pre}?**:
> Your understanding is correct. In Section 6, after Asmp 1, you may assume that any reference to $M_{\theta}$ can be replaced with $P_{pre}$. Asmp 1 helps us answer: “if we had perfect pretraining conditions and exactly learned the target distribution, what are the theoretical characteristics of DPT?” We will clarify this in the text.
>
> However, the proof of Thm 1 is not simply proof-by-definition! PS and DPT are fundamentally different in their operation. The key technical challenge of the proof is showing that the posterior distribution over actions collapses in just the right way to make the two trajectory distributions equivalent.
>
> **Prop 6.3: Is it stating that a pretraining dataset from PPO and a random policy would perform the same?**: Thanks for highlighting this. Prop 6.3 helps explain the following observation. Whether we pretrain with datasets sampled randomly or from an algorithm, DPT has similar behavior. However, some pretraining datasets produce different results (DPT-Exp in Fig 3.a). Prop 6.3 shows pretraining with adaptively collected datasets will *distributionally* lead to the same model. Statistical differences may arise due to finite samples and coverage in practice. This is qualitatively consistent with what we observe in Figs 3.bc and 5.bc.

---

> > ### Comment · Reviewer_CX5S · 2023-08-15
> >
> > Thanks for the thorough rebuttal. Authors have addressed all my concerns, and clarified misunderstandings. I am thus changing my rating.

---

### Author Rebuttal · Authors · 2023-08-09

Dear Reviewers and AC,

Thank you for your detailed and thoughtful reviews. We are delighted to hear that the reviewers found the problem and results interesting and novel (R3a7, pHmJ), and the analyses well-designed and thorough (pHmJ, CX5S). We share excitement about the insights such as the connection to posterior sampling (qnXx, CX5S) and the ability to exploit latent structure from pretraining (R3a7). We also greatly appreciate the feedback and will incorporate it in the revision. Please see the individual responses for answers to specific questions and clarifications. We have also attached a PDF response with new figures addressing certain questions that warranted new experiments. Here are a few key points:
- Reviewers CX5S and qnXx: We have included discussion of how (approximately) optimal actions during offline pretraining can be acquired in many settings, that are a superset of related meta-RL problem settings.
- Reviewer CX5S: We have highlighted major contributions/implications of the work that we believe have not been elucidated in prior work, including how
   - it is not necessary to start with a good RL algorithm to learn one, via supervised pretraining.
   - provably sample-efficient posterior sampling (studied primarily theoretically) can be scaled up via transformers and *supervised* pretraining.
   - learning distributions over task-specific optimal actions (rather than imitating existing algorithms) better leverages data-driven priors.
- Reviewer CX5S: In the attached PDF, we have included additional MAML and Prompt-DT baselines in their respective settings, as well as a MAML visualization.
- Reviewer qnXx: In the attached PDF, we have included a new version of the MiniWorld results with position completely removed and instead images used in the in-context dataset. The pretraining was also scaled up for all methods. We have also included an ablation varying the context size for offline Dark Room (in addition to existing ones in the original submission for offline/online bandits and online Dark Room).

We are happy to answer any additional questions.

---

### Decision · Program_Chairs · 2023-09-21

**Decision:**

Accept (spotlight)

**Comment:**

This paper presents DPT, a new approach for in-context reinforcement learning (ICRL) with transformers. The approach differs from existing ICRL approaches such as Algorithm Distillation in several aspects, such as using the optimal actions as training labels, as well as the flexibility of the in-context dataset. The paper demonstrates that DPT works well experimentally for solving new tasks in-context in several bandits and small RL environments. Theoretically, DPT is shown to implement the posterior sampling algorithm at the minimizer of the population objective.

All reviewers are positive about the contributions of the paper after rebuttal. I believe DPT could be of broad interest to the community as an approach for decision making with transformers. Therefore, I recommend acceptance with spotlight, and congratulate the authors for the nice work.